# WHEN CAN YOU GET AWAY WITH LOW MEMORY ADAM?

## ABSTRACT

Adam is the go-to optimizer for training modern machine learning models, but it requires additional memory to maintain the moving averages of the gradients and their squares. While various low-memory optimizers have been proposed that sometimes match Adam's performance, their lack of reliability has left Adam as the default choice. In this work, we apply a simple layer-wise Signal-to-Noise Ratio (SNR) analysis to quantify when second-moment tensors can be effectively replaced by their means across different dimensions. Our SNR analysis reveals how architecture, training hyperparameters, and dataset properties impact second moment compressibility, naturally leading to *SlimAdam*, a memory-efficient Adam variant. *SlimAdam* compresses second moments along dimensions with high SNR when feasible, and leaves when compression would be detrimental. Through experiments across a diverse set of training tasks, we show that *SlimAdam* matches Adam's performance and stability while saving up to $99\%$ of total second moments ($\sim 50\%$ total memory).

## 1 INTRODUCTION

Adam with weight decay (Loshchilov & Hutter, 2019) has become the standard optimizer choice in modern machine learning, consistently outperforming non-adaptive optimizers such as Stochastic Gradient Descent with momentum (SGD-M). Its success is typically attributed to adapting to the geometry of the landscape by estimating the "effective learning rate" for each parameter using a moving average of the squared gradients. Additionally, this adaptive mechanism makes the optimal learning rate less sensitive to changes in the training recipe. While these factors conspire to make Adam the go-to optimizer for training language models, it requires additional memory beyond SGD-M. It requires storing moving averages of both first and second moments, doubling the optimizer's memory footprint. This memory cost becomes particularly crucial in resource-limited settings, where the memory allocated to the optimizer states could otherwise be used for the model parameters or activations.

To avoid the extra memory footprint of Adam, various low-memory optimizers have been proposed (Shazeer & Stern, 2018; Ginsburg et al., 2020; Anil et al., 2019; Modoranu et al., 2024). These optimizers are a free lunch in some settings – slashing memory usage with no detectable loss in performance (Zhao et al., 2025; Zhang et al., 2025) – but they compromise performance in others (Luo et al., 2023). While the potential benefits of low-memory optimizers are clear, a lack of understanding as to when they will perform well is a major barrier to widespread adoption, as the expense of training modern generative models makes engineers unwilling to take risks such as modifying core components in the training recipe. We argue that a practical low-memory alternative to Adam should exhibit the following properties. First and foremost, it must maintain optimization efficacy, showing no degradation in performance. Additionally, it should preserve Adam's robustness to minor changes in the training hyperparameters. Finally, the low-memory optimizer should immediately work with the same hyperparameter choices as Adam so that users can swap in a low-memory optimizer without major re-tuning.

Figure 1(a) reveals a natural dichotomy in the space of low-memory optimizers: (1) those that yield learning rate sensitivity curves similar to Adam's, and (2) those that deviate substantially, exhibiting major shifts in optimal learning rates and expected training dynamics. The first group comprises Adam-mini and our proposed *SlimAdam*, which are both constructed by replacing individual second moments with their means along specific dimensions, whereas the latter group comprises Lion, SM3, and Adafactor, which are all significantly different algorithms. In this work, we focus on the first category of low-memory optimizers, as they can serve as a drop-in replacement for Adam. Our goal here is to develop a principled framework to help users understand and quantify when these

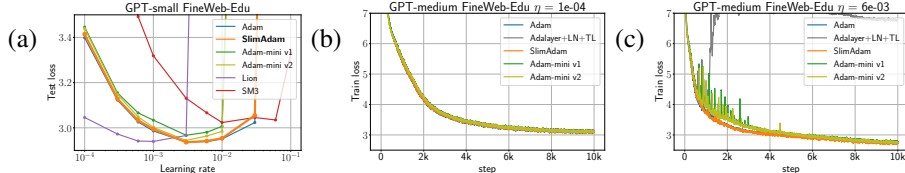

Figure 1: Comparison of common low-memory optimizers on GPT pre-training task using Fineweb-Edu dataset. (a) *SlimAdam* matches Adam's performance with a nearly identical U-shaped loss curve. (b) All low-memory Adam variants exhibit nearly identical training curves at small learning rates. (c) At large learning rates, *SlimAdam* exhibits nearly the same training dynamics as Adam, while other low-memory Adam variants experience training instabilities.

low-memory variants of Adam are appropriate for their problem, thereby improving the reliability of low-memory optimizers and providing deeper insights into Adam's dynamics.

**Contributions:** We propose and study a simple measure of the compressibility of Adam's second-moment memory. By examining the Signal-to-Noise Ratio (SNR) of the second moment tensor in each layer, we quantify when it is safe to replace individual second moments with their means across specific dimensions (such as $\text{fan}_{\text{in}}$ or $\text{fan}_{\text{out}}$). Our SNR analysis reveals that optimal compression strategies vary by layer type and strongly depend on the architecture, training hyperparameters, and dataset properties. For example, key and query second moments of Transformers prefer compression along the $\text{fan}_{\text{in}}$ dimension, as behaviors in the $\text{fan}_{\text{out}}$ dimensions are inconsistent across the multiple heads stacked in that dimension. These compressibility trends are systematic within a specific training domain, but do vary across different domains, reflecting differences in the optimization landscape induced by the model, data, and training objective. For instance, the first MLP layer prefers compression along the $\text{fan}_{\text{out}}$ dimension in GPT pre-training, but along the $\text{fan}_{\text{in}}$ in ViT image classification.

To demonstrate the utility of our analysis, we implement *SlimAdam*, a memory-efficient variant of Adam that utilizes SNR to determine the most efficient dimension for each layer, or selectively leaves layers uncompressed when needed to maintain stability. Since compression trends remain consistent within each task, *SlimAdam* requires configuration once per task, with settings that generalize across scale and dataset variations. By taking an adaptive approach to compression, *SlimAdam* preserves desirable properties of Adam while significantly reducing memory usage. For instance, *SlimAdam* saves $99\%$ of second moments ($\sim 50\%$ total memory) in GPT pre-training, scaling up to 1B parameters. Further, we show that *SlimAdam* matches Adam's performance and robustness to the choice of learning rate.

From a fundamental perspective, our work addresses a critical theoretical question: Does Adam effectively utilize its full second moments during training? Through a careful analysis, we show that many layers can be compressed and pinpoint key compression bottlenecks, such as large vocabulary sizes, large learning rates, and suboptimal initializations. Our investigation also reveals an intriguing observation: the second moment compressibility drastically reduces at large learning rates. For instance, in GPT pre-training, SNR analysis suggests that only $\sim 30\%$ of Adam's second moments are safe to compress at the optimal learning rate. This finding, combined with the success of low memory optimizers, suggests that while only a small fraction of second moments are required to achieve optimal performance, Adam ends up utilizing a significantly large proportion at large learning rates.

**Related Work:** The superiority of Adam is primarily observed in language modeling, with SGD performing comparably to Adam in image classification settings (Zhang et al., 2020). This disparity has motivated several investigations into the unique challenges of language modeling landscapes, with studies identifying several explanations. Zhang et al. (2020); Ahn et al. (2024) argue that the heavy-tailed distribution of the stochastic gradient noise in language modeling cases causes SGD to perform worse than Adam. Pan & Li (2022) attributed Adam's faster convergence to "directional sharpness," which is the curvature along the update direction. Adding to these findings, Zhang et al. (2024) illustrated that the Hessian spectrum varies heavily across parameter blocks, attributing SGD's worse performance to using a single learning rate for all blocks. Further insights come from Kunstner et al. (2024), who found that, in settings with heavy-tailed class imbalance, SGD struggles to decrease loss in infrequent classes, while adaptive optimizers are less sensitive to this imbalance. Zhao et al. (2025) argued that Adam's advantage over SGD in language modeling primarily stems from using per-parameter adaptive learning rates in two specific components—LayerNorm and the final layer.

Several approaches have been proposed to reduce Adam's memory footprint in the past few years. Adafactor (Shazeer & Stern, 2018) approximates the second-moment matrix of a layer using a

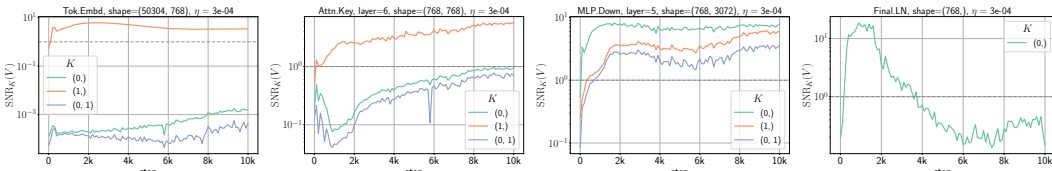

Figure 2: SNR trajectories of selected second-moment blocks of a GPT-small model trained on OpenWebText. Compression dimensions are denoted by: $K = 0$ for fanout, $K = 1$ for fanin, and $K = (0, 1)$ for both dimensions.

moving average of the row and column sums of the squared gradients. SM3 (Anil et al., 2019) groups parameters by tensor dimensions and estimates second moments as the minimum value across relevant group averages. Lion (Chen et al., 2023) is an algorithmically discovered optimizer that only tracks momentum and uses a sign operation to estimate the update. MicroAdam (Modoranu et al., 2024) combines gradient sparsification, quantization, and error feedback to compress optimizer states. Adam-mini (Zhang et al., 2025) assigns adaptive learning rates to block partitions based on the Hessian spectrum at initialization. In Appendix B, we further discuss closely related works in detail.

## 2 NOTATIONS AND PRELIMINARIES

Consider a loss function $L(\boldsymbol{\theta})$ parameterized by parameters $\boldsymbol{\theta}$. For a weight matrix $W \in \mathbb{R}^{\text{fan}_{\text{out}} \times \text{fan}_{\text{in}}}$, let $G_t := \nabla_W L(\boldsymbol{\theta}_t)$ denote its gradient at step $t$. Adam updates these weights using learning rate $\eta_t$ and the moving averages of the first two moments of gradients, denoted by $M_t$ and $V_t$, with coefficients $\beta_1$ and $\beta_2$, respectively. The equations governing the updates are: $M_{t+1} = \beta_1 M_t + (1 - \beta_1)G_t$, $V_{t+1} = \beta_2 V_t + (1 - \beta_2)G_t^2$, and $W_{t+1} = W_t - \eta_t \hat{M}_{t+1}/\sqrt{\hat{V}_{t+1}}+\epsilon$. Here, $\hat{M}_t = M_t/1-\beta_1^t$ and $\hat{V}_t = V_t/1-\beta_2^t$ are the bias-corrected moments and $\epsilon$ is a small scalar used for numerical stability. For our analysis, we generalize Adam to a family of low-memory variants parameterized by layer-specific sharing dimensions. For each layer, we compute an estimate of the second moments by averaging squared gradients across specified dimensions $K$ (fan$_{\text{in}}$, fan$_{\text{out}}$, or both). The difference compared to Adam lies in the second moment update:

$$V_{t+1} = \beta_2 V_t + (1 - \beta_2)\mathbb{E}_K\left[G_t^2\right], \tag{1}$$

where $\mathbb{E}_K[\cdot]$ denotes an average over dimensions $K$. Since Adam's second moment acts as a per-parameter "effective" learning rate, averaging these moments along dimensions $K$ is equivalent to sharing a common learning rate. The above optimizer coincides with Adam when $K = \varnothing$. Another notable limiting case is AdaLayer (Zhao et al., 2025), which maintains one second moment per parameter block. In Section 5, we introduce *SlimAdam*, a special member of the low memory Adam family, where the averaging dimensions $K$ are determined by our SNR analysis.

Throughout this work, we partition second moments using the default parameter partitioning scheme that groups parameters at the granularity of layer components (weights, biases, and attention components), while accounting for special dimensions such as attention heads when interpreting the results. We use $K = 0$ for fan$_{\text{out}}$, $K = 1$ for fan$_{\text{in}}$ and $K = (0, 1)$ to denote sharing along both dimensions.

## 3 SNR ANALYSIS OF ADAM'S SECOND MOMENTS

This section analyzes how effectively Adam's per-parameter second moments can be replaced by their mean along different dimensions (such as fan$_{\text{in}}$, fan$_{\text{out}}$, or both) during training. The feasibility of such a compression depends on how tightly the entries are clustered around their mean value. If entries along a dimension exhibit low variance relative to their mean, they can be effectively represented by a single value. To quantify this concentration of values, we analyze the Signal-to-Noise Ratio (SNR) of the second moments during training. For a second moment matrix $V \in \mathbb{R}^{\text{fan}_{\text{out}} \times \text{fan}_{\text{in}}}$ and specified compression dimensions $K$, $\text{SNR}_K$ is defined as:

$$\text{SNR}_K(V_t) = \mathbb{E}_{K'}\left[\frac{(\mathbb{E}_K[V_t])^2}{\text{Var}_K[V_t]}\right] \tag{2}$$

where $\mathbb{E}_K[\cdot]$ and $\text{Var}_K[\cdot]$ compute the mean and variance along the specified dimensions $K$, while the outer expectation $\mathbb{E}_{K'}[\cdot]$ averages the ratio over the remaining dimensions to obtain a scalar.

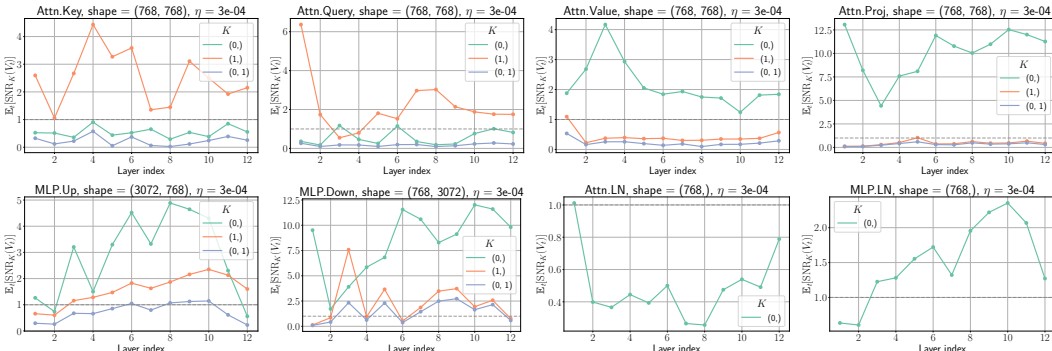

Figure 3: Depth dependence of average SNR values for different second-moment blocks of the GPT-small model trained on OpenWebText. The experimental setup is the same as in Figure 2.

SNR directly quantifies the quality of the compression —when the replacing $V$ with its mean along axis $K$, the relative compression error is given by (see Appendix C.1 for details):

$$\frac{\mathbb{E}_K[\|V - \mathbb{E}_K[V]\|^2]}{\mathbb{E}_K[\|V\|^2]} = \frac{1}{1 + \mathrm{SNR}_K(V)}. \tag{3}$$

Hence, high SNR indicates the mean provides a good approximation with relative error approaching zero, while low SNR suggests significant information is lost during compression. Futhermore, in Appendix C.2, we show that SNR analysis can be viewed as examining Adam's preconditioner through its condition number. As Adam adapts to the local geometry of the loss landscape through its preconditioner, SNR values also serve as a proxy for optimization complexity during training, with lower SNR suggesting higher complexity and a need for per-parameter effective learning rates.

**Compressibility:** Throughout this work, we say that second moments are compressible along dimensions $K$ when $\mathrm{SNR}_K \gtrsim \alpha$, where $\alpha \approx 1$ is a given threshold[1]. When $\mathrm{SNR}_K \gtrsim 1$, the signal dominates the noise, indicating entries can be effectively described by their mean, whereas $\mathrm{SNR}_K \lesssim 1$ suggests that individual entries carry significant information that would be lost when the entries are replaced by their mean. In Section 5, we find that $\alpha = 1$ reliably yields low memory Adam, without any manual tuning.

### 3.1 COMPRESSIBILITY IN DIVERSE TRAINING TASKS

We analyze the evolution of SNR across diverse training tasks (pre-training, fine-tuning, image classification) to uncover fundamental SNR trends. For each setup, experimental details and supplementary results are provided in Appendix D and Appendix F, respectively. We introduce our methodology by examining a representative example. Figure 2 (left) shows SNR trajectories of the second-moment matrix for the Token Embedding layer of a GPT-small model trained on a language pre-training task. These SNR trajectories typically exhibit an early transient phase where their value quickly grows, followed by a late time phase where these values may consistently increase, decrease, or stabilize. We are interested in cases where it is feasible to replace the second moments by their mean throughout training. To this end, we define average SNR as: $\mathbb{E}_\tau[\mathrm{SNR}_K(V_\tau)] = \frac{1}{T}\sum_\tau^T \mathrm{SNR}_K(V_\tau)$, where $\tau$ indexes the training steps at which SNR is measured and $T$ is the total number of measurements. The averaged SNR quantifies compression feasibility along dimensions $K$ throughout training.

#### 3.1.1 LANGUAGE PRE-TRAINING

We analyze GPT-style Transformers (Radford et al., 2019) trained on two language datasets: Open-WebText (Gokaslan et al., 2019) and FineWeb-Edu (Penedo et al., 2024). Figure 2 shows SNR trajectories as a function of the optimization step for selected second-moment blocks of a GPT-small model trained on OpenWebText. Figure 3 presents the depth dependence of the averaged SNR values of different layer types within a standard transformer block. The lack of consistency as to which compression dimension $K$ exhibits higher SNR across different layer types suggests that optimal compression strategies must be customized for each parameter category rather than applying a uniform approach throughout the model. Below, we describe these trends in detail and discuss their implications.

---

[1] For random Gaussian gradients, $\mathrm{SNR}_K > 1/2$ indicates compression feasibility (see Appendix C.3).

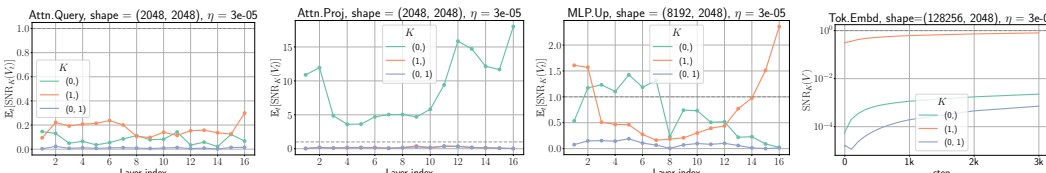

Figure 4: SNR trends for selected layers of pre-trained Llama 3.2 1B fine-tuned on Alpaca dataset. (for full results, see Appendix F.2)

Token Embedding and Language Modeling Head (LM Head[2]) second moments exhibit lower compressibility (low SNR) along the token dimension (vocabulary dimension) while favoring compression (high SNR) along the embedding dimension. This pattern suggests that the subset of the parameter matrix corresponding to each token in the vocabulary evolves at its own pace during training, thereby requiring its own learning rate. These findings align with recent studies (Zhang et al., 2025; Zhao et al., 2025) that advise against compressing the token embedding and LM Head matrices in language modeling. Our SNR analysis extends this understanding by revealing that this lower compressibility is specific only to the token dimension and not the entire matrix.

Attention key and query second moments consistently show lower compressibility along the $\text{fan}_\text{out}$ dimension, where multiple heads are stacked, suggesting that each attention head requires its own effective learning rate. (Zhang et al., 2025) reached similar conclusions through an independent Hessian-based analysis, corroborating our findings. On the other hand, attention value and projection second show an opposite trend, with higher compressibility along the $\text{fan}_\text{out}$ dimension as compared to the $\text{fan}_\text{in}$ dimension. For projection layers, lower compressibility along the $\text{fan}_\text{in}$ dimension (where heads are stacked) is intuitive, as the parameters corresponding to each attention head are intended to evolve independently during training. However, for the same reason, the higher compressibility of second moments in the value layer along the head-stacked dimension is unexpected. Intuitions aside, from an absolute magnitude perspective, values and projection layers show higher averaged SNR values along the preferred dimension than keys and queries, indicating greater overall compressibility.

Interestingly, by a similar magnitude argument, MLP second moments exhibit greater compressibility than attention keys and queries. While in general MLP second moments exhibit higher SNR values along the output dimension ($\text{fan}_\text{out}$), for some layer indices the second moment can also exhibit higher compressibility along the input dimension ($\text{fan}_\text{in}$) or even both dimensions simultaneously.

LayerNorm components show different SNR trends depending on their position in the network. The SNR values of the attention LayerNorms and final LayerNorm typically exhibit a sharp decline after an initial increase, suggesting incompressibility. In contrast, MLP LayerNorms maintain consistently high SNR values throughout training, indicating their second moments can be effectively compressed. We validate the robustness of these results in Appendix F.1 by observing similar trends in a larger model and on a different dataset (FineWeb-edu).

### 3.1.2 LANGUAGE FINE-TUNING

Next, we examine second-moment compressibility during fine-tuning with Llama-3.2 (Team, 2024) on the Alpaca dataset (Taori et al., 2023). Figure 4 shows the SNR trends of selected layers, which reveal layer-wise patterns with subtle distinctions from GPT pre-training. We find lower SNR values across all layers during fine-tuning, suggesting lower compressibility in general in this experimental setting. This is particularly pronounced in the attention layer, where key and query second moments exhibit SNR values well below $1.0$. While attention value and projection second moments maintain an SNR value above $1.0$ along $\text{fan}_\text{out}$ dimension, these values are notably smaller than those observed during GPT pre-training. MLP layers display variable SNR patterns. The first two MLP layers (MLP.Up and MLP.Gate) show sporadic compressibility (SNR $\gtrsim 1$) at certain depths, but without consistently favoring either input or output dimension compression. In comparison, the output MLP layer (MLP.Down) consistently maintains a high SNR value (SNR $\gtrsim 1$) across depths, exhibiting higher compressibility along the $\text{fan}_\text{out}$ dimension. Attention and MLP RMSNorms show consistently low SNR values across layers, while the final RMSNorm's SNR gradually increases during training, eventually exceeding $1.0$. The token embeddings show reduced SNR values even along the embedding dimension, possibly due to a larger vocabulary relative to the embedding dimension for the Llama model than the GPT-small model.

---

[2]We use weight tying, meaning that the Token Embedding and LM Head share parameters and moments.

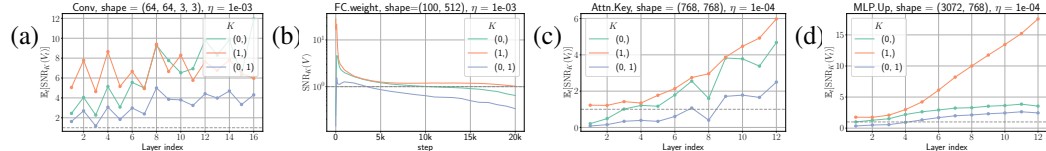

Figure 5: SNR trends of selected layers of (a, b) ResNet-18 and (c, d) 12-layer ViT trained on CIFAR-100. For detailed results, see Appendix F.3.

### 3.1.3 ResNet Image Classification

Compared to language pre-training and fine-tuning settings, the second moments of ResNets trained on CIFAR-100 and CIFAR-10 (Figure 5(a, b) and Appendix F.3) exhibit high SNR values. These SNR values suggest high second-moment compressibility across layers. In particular, the intermediate convolutional layers show exceptionally high SNR values across both $\text{fan}_{\text{in}}$ and $\text{fan}_{\text{out}}$ dimensions, with an increasing trend as a function of depth. By comparison, the first and last layers behave differently. The first convolutional layer resists compression along the $\text{fan}_{\text{out}}$ dimension (shown in Figure 24, Appendix F.3), while the final layer exhibits SNR values close to 1.0 that decreases late in training.

### 3.1.4 ViT Image Classification

Next, we analyze Vision Transformers (ViTs) (Dosovitskiy et al., 2021), with a GPT-style Transformer adapted for image classification. Figure 5(c, d) shows that ViTs trained on CIFAR-100 exhibit SNR trends combining characteristics from both ResNet and GPT pre-training. The attention moments maintain GPT-like SNR trends but with higher SNR values. The keys and query second moments favor $\text{fan}_{\text{in}}$ compression, while values and projections prefer $\text{fan}_{\text{out}}$ dimension. These attention components exhibit higher SNR values than GPT pre-training, with the averaged SNR increasing with depth for most layers. Unlike GPT pre-training, the first MLP layer (MLP.Up) favors $\text{fan}_{\text{in}}$ compression instead of $\text{fan}_{\text{out}}$. This suggests that this layer type's compression behavior is training task-dependent. By comparison, the second layer (MLP.Down) maintains GPT-like $\text{fan}_{\text{out}}$ preference and exhibits high SNR values along both dimensions. Similar to ResNet's first convolution layer, ViT's patch embedding layer favors $\text{fan}_{\text{in}}$ compression. Meanwhile, the classification layer maintains SNR values close to 1.0 without consistent preference toward a particular compression dimension. Notably, all LayerNorm components display surprisingly high SNR values, suggesting high compressibility.

### 3.2 Compressibility Trends Across Training Tasks

SNR analysis revealed several consistent compressibility trends and some task-specific behaviors. Table 1 summarizes the preferred compressibility dimension by layer type, which we discuss below.

**Attention:** Key and query second moments consistently exhibit higher compressibility along $\text{fan}_{\text{in}}$ dimension while showing lower compressibility along $\text{fan}_{\text{out}}$ (head-stacked dimension). By comparison, values and projection second moments display an opposite trend, exhibiting higher compressibility along $\text{fan}_{\text{out}}$ dimension. Moreover, value and projection layers generally exhibit higher SNR values than key and query layers, suggesting higher overall compressibility. These trends persist across training tasks (GPT pre-training, Llama fine-tuning, and ViT image classification), suggesting these trends are intrinsic to the attention mechanism. However, the compressibility strength varies across training tasks, with ViT showing overall higher SNR values than GPT pre-training and fine-tuning exhibiting notably lower SNR values.

Table 1: Summary of preferred compression dimensions by layer type. Compression dimensions marked with $\star$ show inconsistent trends across training tasks. Compared to prior work, we report different trends for attention value, projection, and normalization layers (see Appendix B for details).

| Layer Type | $\mathbf{K}^{*}$ | Layer Type | $\mathbf{K}^{*}$ |
|---|---|---|---|
| *Attention* | | *Special Layers* | |
| Key & Query | $\text{fan}_{\text{in}}$ | Token Embedding | $\text{fan}_{\text{out}}$ |
| Value & Projection | $\text{fan}_{\text{out}}$ | Language Modeling Head | $\text{fan}_{\text{in}}$ |
| *MLP Layers* | | Vision First Layer | $\text{fan}_{\text{in}}$ |
| First layer (Up) | $\text{fan}_{\text{out}}^{\star}$ | Vision Classification Head | $\text{fan}_{\text{in}}^{\star}$ |
| Middle layer (Gate) | $\text{fan}_{\text{out}}^{\star}$ | *Normalization Layers* | $-$ |
| Last layer (Down) | $\text{fan}_{\text{out}}$ | | |

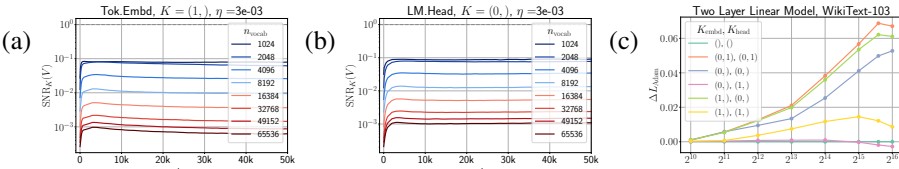

Figure 6: (a) SNR trajectories of the linear head of the simplified two-layer model with varying vocabulary sizes. For details, see Appendix I. (b) Test loss gap $\Delta L_{\text{Adam}} = L_{(K_{\text{embd}}, K_{\text{head}})} - L_{\text{Adam}}$ of the linear model trained with Adam with shared second moments across dimensions $(K_{\text{embd}}, K_{\text{head}})$.

**MLPs:** Our GPT and ViT models share identical MLP blocks with two layers (MLP.Up and MLP.Down). The first layer shows task-dependent trends, with higher fan$_{\text{out}}$ compressibility in the language pre-training and fan$_{\text{in}}$ in ViT image classification. The second layer (MLP.Down), consistently exhibits higher fan$_{\text{out}}$ compressibility across both settings. The pre-trained Llama model uses three layers in the MLP block (Up, Down, Gate). The first two layers (Up, Gate) show inconsistent compressibility trends, whereas the output layer (Down) exhibits higher fan$_{\text{out}}$ compressibility similar to the GPT setting.

**First and Last layer:** In language models, Token Embedding and LM Head exhibit lower SNR values along the token dimension, while maintaining higher values along the embedding dimension. In image classification, the first layers exhibit higher fan$_{\text{in}}$ compressibility, while classification heads show inconsistent compression trends but maintain overall higher SNR values. Overall, image classification models exhibit substantially higher compressibility than language models.

**Normalization layers:** These layers show domain-specific compressibility trends. Language models exhibit lower LayerNorm compressibility, while both BatchNorm and LayerNorm in vision models maintain higher compressibility throughout training. Due to their high variability and minimal contribution to the overall memory usage, we advise against compressing them.

## 4 FACTORS INFLUENCING COMPRESSIBILITY

Our earlier analysis revealed various consistent SNR trends across training tasks. Here, we conduct experiments to analyze the effect of initialization, dataset properties, and learning rate on these trends.

### 4.1 INCOMPRESSIBILITY UNDER HEAVY-TAILED DISTRIBUTIONS

In the previous section, we observed that language models exhibit very low SNR along the token dimensions. This suggests that individual tokens require their own learning rates, as their gradients evolve at different paces. To better understand this phenomenon, we investigate how token frequency distribution influences compressibility. We examine a simplified two-layer model, solely consisting of a token embedding matrix and a linear head. We train the model on the WikiText-103 dataset (Merity et al., 2017), tokenized using BPE tokenizer (Gage, 1994) with varying vocabulary sizes. By progressively reducing the vocabulary size, we systematically remove rare tokens to control the tail of the token distribution. Figure 6(a, b) shows that SNR values along the token dimension of both layers decrease substantially as vocabulary size increases, suggesting lower compressibility.

We then analyze how large vocabulary sizes affect performance by training the model using Adam with shared second moments (Equation (1)) along dimensions $(K_{\text{embd}}, K_{\text{head}})$. Figure 6(c) shows the loss gap between the above optimizer and standard Adam, defined as $\Delta L_{\text{Adam}} = L_{(K_{\text{embd}}, K_{\text{head}})} - L_{\text{Adam}}$. For large vocabularies, compression is only effective along embedding dimensions, while token-dimension compression degrades performance. In contrast, small vocabularies permit compression along both dimensions. These findings extend the work of Kunstner et al. (2024), which showed that Adam outperforms SGD on language tasks by making faster progress on rare tokens. Our analysis suggests that the apparent advantage of Adam in language modeling might stem in large part from allowing individual second moments to each token in the vocabulary.

### 4.2 LARGE LEARNING RATES REDUCE COMPRESSIBILITY

In this section, we analyze how increasing the learning rate affects averaged SNR values and thereby compression feasibility. Figure 7(a, b) shows that increasing the learning rate consistently reduces SNR values across layers (see Appendix G.1 for full results). For clarity, we focus on the preferred SNR compression dimension for each layer type. This decline in averaged SNR values suggests that higher learning rates cause training to explore regions of parameter space where the gradient distribution contains more outliers, thereby reducing SNR values. In Appendix G.2, we show that

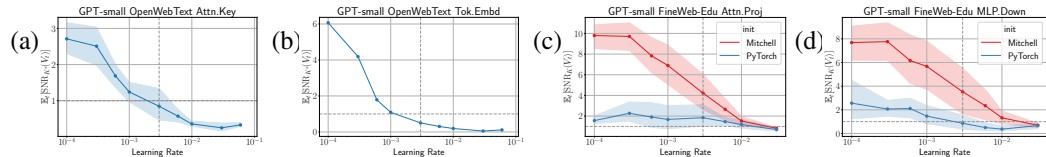

Figure 7: (a, b) The effect of learning rate on the averaged SNR values of selected layer types of a GPT-small model trained on OpenWebText. For each layer type, we select the compression dimension $K^*$ with the highest SNR. The shaded region around the mean trend shows the variation across depth. (c, d) The effect of initialization on SNR trends of GPT-small trained on FineWeb-Edu.

average SNR trends do not vary extensively with batch size and learning rate remaining the primary factor reducing compressibility. Based on the effect of increasing the learning rate on SNR values, we classify layer types into two categories:

1. *Layers that have lower compressibility (SNR $\lesssim$ 1) at the optimal learning rate:* Token Embedding/LM Head, LayerNorm, Attention keys, queries, first MLP layer (MLP.Up).

2. *Layers that exhibit higher compressibility (SNR $\gtrsim$ 1) at the optimal learning rate:* Attention values and projections and the second MLP layer (MLP.Down).

Pre-trained Llama and ViT models show similar results, while ResNets remain compressible even at very high learning rates. In Section 5, we quantify these architectural differences in compressibility.

### 4.3 Effect of Initialization on Compressibility

Next, we examine the effect of initialization schemes on SNR trends. We compare Mitchell initialization (Groeneveld et al., 2024) used in Section 3 against PyTorch's default initialization scheme. A key feature of Mitchell initialization is that it scales the variance of layers that add to the residual stream (Attn.Proj and MLP.Down) with a factor of $1/$depth. Figure 7(c, d) and Figure 30 in Appendix H show that Mitchell initialization leads to higher SNR values compared to the default PyTorch initialization across layers of the GPT-small model. In particular, Attn.Proj and MLP.Down layers show significantly higher SNR values. These exceptionally high SNR values provide empirical support for the $1/$depth scaling in Mitchell initialization. As Adam's second moments adapt to the landscape geometry, these findings indicate that SNR analysis can serve as a proxy for evaluating initialization schemes by determining ones with higher SNR values.

## 5 Building A Low-Memory Adam Variant

Leveraging insights from our comprehensive analysis of SNR trends presented in previous sections, we now introduce *SlimAdam* — a memory-efficient Adam variant that preserves Adam's performance and stability through SNR-guided compression. In a nutshell, *SlimAdam* compresses matrix-like second moments along the dimension with the highest SNR when it is above a threshold and leaves vector-like second moments uncompressed due to their high variability and minimal effect on the overall memory. Our implementation consists of three steps (see Figure 8 for an overview):

**Step 1:** First, we collect layer-wise SNR statistics using a small proxy model with a $10\times$ smaller learning rate than optimal.

**Step 2:** Next, we identify the compression dimension $K^*$ for each layer type with the highest SNR, i.e., $K^* = \arg\max_K \mathbb{E}_\tau[\text{SNR}_K(V^{(l)})]$. We compress a layer only if $\mathbb{E}_\tau[\text{SNR}_{K^*}(V^{(l)})]$ is above a given cutoff. Otherwise, no compression is applied and Slimadam reverts to full Adam for this layer.

**Step 3:** Finally, we train the target model using Adam with shared second moments (Equation (1)) along these compression dimensions $K^*$ determined in Step 2.

For new training tasks, we recommend deriving compression rules, whereas for well studied setups, such as GPT pre-training, our prec-computed rules in Table 1 can be directly used. The full algorithm is detailed in Appendix E.1.

**The Superior Stability of SlimAdam:** Figure 1(b, c) shows that *SlimAdam* exhibits more stable training dynamics at large learning rates compared, unlike other low-memory Adam variants. By comparison, at small learning rates, all low-memory Adam variants perform equally well. While these alternatives show large training instabilities at Adam's optimal learning rate, *SlimAdam* maintains nearly identical training dynamics as Adam. This difference in stability is expected, as for Adam

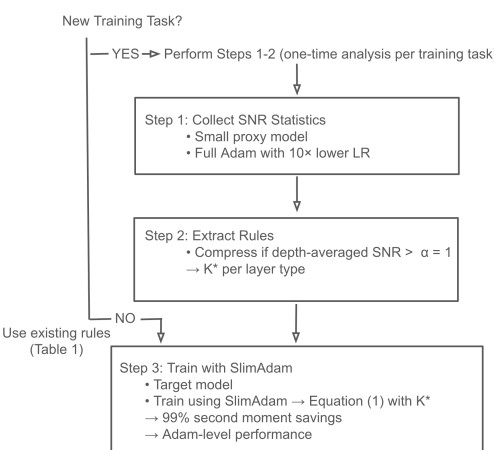

Figure 8: Workflow for building low-memory optimizers using SNR-guided compression.

variants, the pre-conditioner $P^{-1} = 1/\sqrt{V}$ directly affects the local instability threshold (Cohen et al., 2022; Kalra & Barkeshli, 2024), and compressing the "correct" dimensions as guided by our SNR analysis is crucial for maintaining both stability and performance at large learning rates.

**Efficient Analysis with Small Proxy Models:** We find that depth-averaged SNR $\frac{1}{\text{depth}} \sum_{l=1}^{\text{depth}} \mathbb{E}_\tau[\text{SNR}_K(V^{(l)})]$ yields consistent compression dimensions across model sizes. For example, a 4-layer GPT model with embedding dimension $n_{\text{embd}} = 256$ yields the same optimal compression dimensions as a 24-layer model with $n_{\text{embd}} = 1024$, matching those in Table 1. This consistency allows using a smaller proxy model to identify compression dimensions for larger target models. Figure 32 in Appendix J verifies that *SlimAdam* with depth-averaged SNR derived rules yield the same performance in GPT pre-training.

**The Importance of Computing SNR at Small Learning Rates:** SNR-predicted compressibility primarily depends on the learning rate used to train the proxy model and the SNR cutoff, with distinct patterns across architectures (top panel, Figure 9). A priori, one might assume that performing the SNR analysis at the optimal learning rate is also optimal for determining compression rules. However, surprisingly, for Transformer-based models (GPT, Llama, and ViT), we find high compressibility of $\sim$ 99% (SNR cutoff of 1.0) if analyzed at relatively small learning rates but that these savings reduce to $\sim$ 30% at large learning rates. As shown in Figure 9 (bottom panel), for GPT, ViT, and ResNets, deriving compression rules at $10\times$ lower learning rates than optimal can enable *SlimAdam* to achieve Adam-level performance and stability while saving $\sim$ 99% second moments ($\sim$ 50% total memory). In Figure 14, we show that *SlimAdam* continues to match Adam's performance and stability at 1B scale.

## 6 DISCUSSION

Our computationally efficient SNR analysis independently confirms and extends several findings from prior work while overcoming their limitations. Zhang et al. (2025) used Hessian-based analysis of small models to construct a low-memory optimizer and then applied these rules to larger models, assuming transferability. A primary advantage of our approach is that we can directly analyze models of any scale without requiring expensive Hessian computations. Similarly, Zhao et al. (2025)'s extensive ablation studies showed that Adam's advantage over SGD in language modeling primarily stems from maintaining per-parameter second moments for two components: LM Head and LayerNorm. Our SNR analysis naturally uncovers these same trends and shows that for LM Head and Token Embedding, this aversion to compression is specific only to the token dimension.

Our SNR analysis can also serve as a standalone diagnostic tool. SNR values serve as a proxy for learning complexity within each layer, with lower SNR indicating higher complexity. This insight naturally reveals regions of model architecture that could benefit from improvements. For instance, low SNR values observed in token embeddings or language model heads suggest these components might benefit from more sophisticated designs or specialized optimizer rules. SNR analysis also enables a quanti-

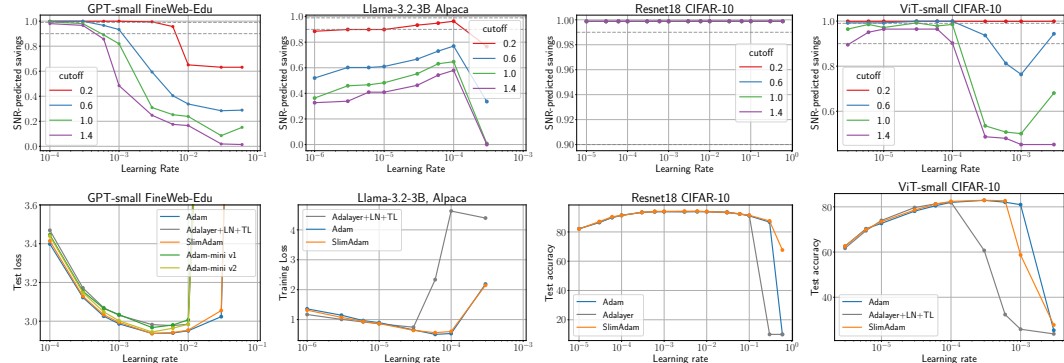

Figure 9: (Top) Fraction of reducible second moments (relative to Adam) across learning rate and SNR cutoff, as predicted by SNR analysis. (Bottom) Performance comparison across learning rates between SlimAdam (with rules derived with 10x smaller learning rate than optimal) and baselines: Adam, AdaLayer, and Adam-mini (for details, see Appendix B). SlimAdam achieves Adam-level performance and stability while significantly reducing memory usage across all configurations. Appendix E.2 shows that SNR cutoff and learning rate does not affect *SlimAdam*'s performance.

tative evaluation of the effectiveness of initialization schemes; see Section 4.3 where lower SNRs under PyTorch's default compared with the "Mitchell" initialization suggest the former's suboptimality.

Based on *SlimAdam*'s success in the GPT pre-training regime, we posit the following "implicit bias" of the Adam optimizer. Without any specific intervention during training, Adam's full second moment tensors end up populated with incompressible state regardless of whether the optimization problem— the architecture, dataset, and/or objective—actually requires this much flexibility. Only through SNR analysis at small learning rates, where we can avoid artifacts that emerge when training Adam at large learning rates, are we actually able to capture these latent fundamental compression rules the that optimization problem admits. Since only a small fraction of second moments are required for optimal performance, we interpret our results as evidence of an inherent bias of Adam to utilize whatever state capacity is provided, even if it is not strictly necessary for optimal training performance.

In conclusion, we present a principled framework to analyze when second moments can be effectively replaced with their means, naturally leading to *SlimAdam*, a practical low-memory Adam variant which maintains performance and stability while enjoying significant memory savings. A key limitation of SNR analysis is that it's based on an Adam run (which depends on hyperparameters) and does not guarantee that other low-memory optimizers with even greater memory savings could exist. While conducted up to 1B scale, we hope that our work furthers understanding of when low memory optimizers are safe to use in practice while deepening our fundamental understanding of how architecture, data, and optimizer design interact.

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

## A    LLM USAGE

LLMs were used for editing and condensing paragraphs to comply with the page limit restriction. We have verified that these edits do not change the intended message or result in any way.

## B    DETAILED COMPARISON WITH OTHER LOW-MEMORY OPTIMIZERS

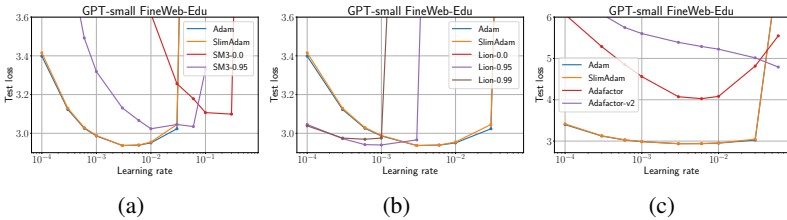

|  |  |  |
|---|---|---|
| (a) | (b) | (c) |

Figure 10: Comparison of *SlimAdam* with different optimizers on GPT pre-training using Fineweb-Edu dataset.

**Adam-mini**: Zhang et al. (2025) introduced Adam-mini, which assigns adaptive learning rates to block partitions based on the Hessian spectrum at initialization. The initial release, Adam-mini v1.0.4 (referred to as Adam-mini v1), uses PyTorch's default block partitioning with two key modifications: (1) individual second moments are assigned to each parameter in the Token Embedding and LM Head, and (2) individual second moments are assigned to each key and query attention head. In a recent update, Adam-mini v1.1.1 (referred to as Adam-mini v2) revises this approach by assigning one second moment per output neuron in each layer, with two exceptions: (1) each key and query attention head receives its own second moment, and (2) each token dimension in the Token Embedding and LM Head receives its own second moment. LayerNorms are always compressed.

Our SNR analysis identifies similar compression rules to Adam-mini, but with two key differences. First, Adam-mini assigns one second moment to every output neuron of attention values, projection, and MLPs. In our convention, it amounts to $fan_{in}$ compression. In comparison, our SNR analysis suggests that $fan_{out}$ compression is more appropriate for these layers. The second difference relates to LayerNorm parameters. While Adam-mini compresses these by default, our SNR analysis indicates that LayerNorm second moments show aversion to compression. We attribute *SlimAdam*'s superior learning rate stability to its identification of these more appropriate compression dimensions.

**AdaLayer:** Zhao et al. (2025) found that Adam's superior performance over SGD in language modeling primarily comes from using per-parameter adaptive learning rates in just two components: LayerNorm and the LM Head. All other layers can be trained with SGD. Following their naming convention, we use AdaLayer to refer to Adam with one second moment per weight/bias, and 'AdaLayer+LN+TN' to denote AdaLayer with per-parameter second moments for LayerNorm and final layer parameters.

While our SNR analysis supports their findings about Token Embedding/LM Head and LayerNorm second moments, we find that AdaLayer+LN+TN underperforms Adam and *SlimAdam*, using 2% of Adam's second moment, closely matches Adam's performance and stability.

**SM3:** Anil et al. (2019) grouped parameters into sets based on similarity, such that each parameter can belong to multiple sets. Then, it maintains a moving average of the maximum of squared moments for each set and approximates a second-moment entry using the minimum value across different sets it belongs to. We use the implementation from Enealor (2020) with momentum $= 0.9$ and $\beta \in \{0.0, 0.95\}$. Figure 10(a) compares SM3 performance with different $\beta$ values on the GPT pre-training task. We observe that $\beta = 0.95$ performs better for GPT pre-training. We use this optimal $\beta$ value in the comparisons shown in Figure 1.

**Lion:** Chen et al. (2023) algorithmically discovered an optimizer that only tracks momentum and uses the sign operation to determine update directions. For the GPT-small experiment, we found that $\beta_2 = 0.95$ performs best when keeping $\beta_1 = 0.9$ fixed, as shown in Figure 10(b). Similar to other optimizers, we use a weight decay strength of $\lambda = 0.1$ and a gradient clipping threshold of $1.0$.

**Adafactor:** Shazeer & Stern (2018) approximated the second-moment matrix of a layer using a moving average of the row and column sums of the squared gradients. We evaluate two implementations: (1) the PyTorch implementation, which does not use a moving average of updates (referred to as Adafactor) and (2) the implementation by Facebook Research (2023), which incorporates the moving average of updates (referred to as Adafactor v2). For both variants, we maintain the same learning rate schedule used in our default experiments. For Adafactor v2, this requires setting `relative_step=False`. As shown in Figure 10(c), both Adafactor variants perform significantly worse than Adam. Due to this performance gap, we exclude these results from Figure 1.

## C    THEORETICAL ANALYSIS OF SIGNAL-TO-NOISE RATIO

Different variants of SNR have been utilized in prior works. For instance, (Schaul et al., 2013) used per-gradient SNR to adaptively set the learning rate, Xu et al. (2024) used SNR to construct a pre-conditioner at initialization, whereas Xiang et al. (2023) used SNR to search over architectures. By comparison, we use SNR to analyze the compression of second moments.

This section theoretically analyzes the SNR metric, examining its fundamental properties and practical implications for adaptive optimization.

### C.1    THE CONNECTION BETWEEN SNR AND RELATIVE COMPRESSION ERROR

In this section, we establish a relationship between SNR and relative compression error. Consider a second moment vector $\mathbf{v} \in \mathbb{R}^n$ with mean $\mu = \frac{1}{n} \sum_{i=1}^n v_i$, variance $\sigma^2 = \frac{1}{n} \sum_{i=1}^n (v_i - \mu)^2$, and SNR$= \frac{\mu^2}{\sigma^2}$. Then, the relative compression error is given by:

$$\frac{\|v - \mu\|^2}{\|v\|^2} = \frac{n\sigma^2}{n(\sigma^2 + \mu^2)} = \frac{1}{1 + SNR}. \tag{4}$$

A high SNR indicates that the mean provides a good approximation with relative error approaching zero, while low SNR suggests significant information is lost during compression.

### C.2    THE CONNECTION BETWEEN SNR AND PRE-CONDITIONING

For adaptive optimizers with pre-conditioner $P$, the dynamics is governed by the pre-conditioned Hessian $P^{-1}H$ (Cohen et al., 2022; Kalra & Barkeshli, 2024), where $P \propto \sqrt{V}$ for Adam. The condition number of the problem is bounded by:

$$\kappa(P^{-1}H) \leq \kappa(P^{-1})\kappa(H), \tag{5}$$

suggesting that the condition number of the preconditioner $\kappa(P^{-1})$ directly influences the overall conditioning. We now connect SNR with the condition number $\kappa(P^{-1})$.

Consider a second moment vector $\mathbf{v} \in \mathbb{R}^n$ with mean $\mu = \frac{1}{n} \sum_{i=1}^n v_i$, variance $\sigma^2 = \frac{1}{n} \sum_{i=1}^n (v_i - \mu)^2$, and SNR $= \frac{\mu^2}{\sigma^2}$. Then, the condition number of Adam's preconditioner is:

$$\kappa(P^{-1}) = \frac{\lambda_{\max}(P^{-1})}{\lambda_{\min}(P^{-1})} = \frac{\min_i^n \sqrt{v_i}}{\max_i^n \sqrt{v_i}}. \tag{6}$$

Both the condition number $\kappa(P^{-1})$ and the SNR measure the dispersion of the second moment distribution.

**High SNR regime:** When SNR is large, second moments concentrate around the mean, resulting in $\kappa(P^{-1}) \approx 1$. In this case, the preconditioner uniformly scales the Hessian by a scalar value, suggesting that a single second moment suffices.

**Low SNR regime:** When SNR is low, $\kappa(P^{-1})$ is small, and replacing second moments with their mean won't perform the required preconditioning.

The above analysis suggests that that SNR analysis can be viewed as examining Adam's preconditioner throughout training.

### C.3 SNR ANALYSIS FOR GAUSSIAN GRADIENTS

In this section, we analyze the SNR metric for random, iid Gaussian distributed gradients. Consider a gradient matrix $G \in \mathbb{R}^{n \times n}$ with elements $G_{ij}$ sampled from $\mathcal{N}(0, \sigma^2)$. Let $V = G^2$ denote the element-wise squared gradient matrix. Then, the expectation of the mean and variance along column $j$ is:

$$\mathbb{E}[V_i] = \mathbb{E}\left[\frac{1}{n} \sum_{j=1}^n V_{ij}\right] = \frac{1}{n} \sum_{j=1}^n \mathbb{E}[G_{ij}^2] = \sigma^2.$$

$$\mathbb{E}\left[\frac{1}{n} \sum_{j=1}^n (V_{ij} - \mathbb{E}[V_i])^2\right] = \frac{1}{n} \sum_{j=1}^n \left[\mathbb{E}[G_{ij}^4] - \mathbb{E}[G_{ij}]^2\right] = 3\sigma^4 - \sigma^4 = 2\sigma^4.$$

This yields SNR $= 1/2$ for iid Gaussian gradients irrespective of matrix dimension. We numerically verify this result in Figure 11. In real-world scenarios, gradients follow complex distributions, often exhibiting long tails that defy iid Gaussian assumptions. In our experiments, we found that a more stringent cutoff of 1 works better.

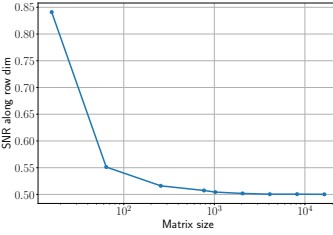

Figure 11: SNR values along the row dimension for iid Gaussian distributed gradients.

## D EXPERIMENTAL DETAILS

**SNR measurement:** We measured SNR values at regular intervals throughout training: every 100 step for the first 1000 steps, then every 1000 step thereafter. For determining the *SlimAdam* rules, we deliberately exclude frequent early-training measurements to prevent biasing the averaged SNR towards initial SNR values.

### D.1 LANGUAGE PRE-TRAINING

**Model and Datasets:** We train GPT-style models (Radford et al., 2019) using a codebase based on NanoGPT (Karpathy, 2022) on two language modeling datasets: OpenWebText (Gokaslan et al.,

2019) and 10B token subset of FineWeb-Edu (Penedo et al., 2024). The datasets are tokenized using the GPT tokenizer with a vocabulary size $n_{\text{vocab}} = 50,304$. The models are trained with a context length of $T_n = 1024$. We use $n_{\text{layers}}$ to denote the number of layers, $n_{\text{heads}}$ to denote the number of heads, and $d_{\text{model}}$ to denote the embedding dimension. We consider three model configurations, as summarized in the table below.

Table 2: GPT model configurations and parameter counts.

| Model | $n_{\text{layers}}$ | $n_{\text{heads}}$ | $d_{\text{model}}$ | Parameters | Tokens |
|---|---|---|---|---|---|
| GPT-small | 12 | 12 | 768 | 100M | 10B |
| GPT-medium | 24 | 16 | 1024 | 350M | 10B |
| GPT-large | 36 | 25 | 1600 | 1B | 20B |

All models have an MLP upscaling factor of 4, learnable positional embedding, and weight tying, without biases.

**Initialization:** Unless specified, we consider the Mitchell initialization (Groeneveld et al., 2024): For standard layers, the weights are initialized using a normal distribution $\mathcal{N}(0, 0.02^2)$, while residual projection layers (attention and MLP projections) use a scaled normal distribution $\mathcal{N}(0, 0.02^2/2n_{\text{layers}})$. In Section 4.3, we use PyTorch's default uniform initialization: $\mathcal{U}(-\frac{1}{\sqrt{\text{fanin}}}, \frac{1}{\sqrt{\text{fanin}}})$.

**Training:** The training uses a micro-batch size of 32 with 40 gradient accumulation steps, resulting in an effective batch size of $B = 1,280$. Small and medium models are trained for 10B tokens, whereas the 1B model is trained for 20B tokens (Chinchilla optimal). We use the following Adam hyperparameters: $\beta_1 = 0.9$, $\beta_2 = 0.95$, $\epsilon = 10^{-8}$, and weight decay strength $\lambda = 0.1$. The learning rate is linearly increased from zero to a target learning rate $\eta$ in $T_{\text{wrm}} = 2048$ steps, followed by cosine decay to $\eta_{\text{min}} = \eta/10.0$. Gradients are clipped at a maximum norm of 1.0.

### D.2 LINEAR MODEL TRAINED ON WIKITEXT

**Model Architecture:** We consider a two-layer linear model composed of an embedding layer followed by a language model head, trained on WikiText-103 (Merity et al., 2017). The dataset is tokenized using BPE tokenization (Gage, 1994; Sennrich et al., 2016) with different vocabulary sizes $V \in \{1024, 2048, 4096, 8192, 16384, 32768, 49152, 65536\}$. The embedding dimension is set to $d_{\text{model}} = 768$ and a context length of $T_n = 128$ is considered.

**Initialization:** The embedding parameters are initialized using a truncated normal distribution $\mathcal{N}(0, 1)$, while the language model head uses a truncated normal distribution $\mathcal{N}(0, 1/\text{fan}_{\text{in}})$.

**Training:** The training consists of one epoch with a batch size $B = 16$. The model is trained using Adam variants with hyperparameters $\beta_1 = 0.9$, $\beta_2 = 0.999$, $\epsilon = 10^{-8}$, and weight decay strength $\lambda = 10^{-4}$. The learning rate follows a schedule with linear warmup from zero to $\eta$ over $T_{\text{wrm}} = 2048$ steps, followed by cosine decay to $\eta_{\text{min}} = \eta/10.0$. The optimal target learning rate is found by scanning the set $\{1e\text{-}4, 3e\text{-}4, 6e\text{-}4, 1e\text{-}3, 3e\text{-}3\}$.

### D.3 LANGUAGE FINE-TUNING

**Model and Datasets:** We consider pre-trained Llama-3.2 models (Team, 2024) and fine-tune them on the Alpaca dataset (Taori et al., 2023) using the torchtune library (torchtune maintainers & contributors, 2024).

**Fine-tuning:** We finetune the models for 3 epochs using a batch size $B = 16$, optimizer hyperparameters $\beta_1 = 0.9$, $\beta_2 = 0.999$, $\epsilon = 10^{-8}$ and weight decay strength $\lambda = 0.1$.

### D.4 IMAGE CLASSIFICATION

**Model and Datasets:** We train ResNet (He et al., 2015) and ViT (Dosovitskiy et al., 2021) models on CIFAR-10 and CIFAR-100 datasets (Krizhevsky et al., 2009) with random crop and horizontal flip augmentations.

**ResNet:** We consider the standard ResNet-18 architecture with batch normalization.

**ViT**: We consider Vision Transformers (Dosovitskiy et al., 2021), with GPT-like architecture adapted for image classification using patch embeddings and a special class token. We consider two model

configurations: ViT-mini ($n_{\text{layers}} = 6$ layers, $n_{\text{heads}} = 12$ heads, embedding dimension $d_{\text{model}} = 768$) and ViT-small ($n_{\text{layers}} = 12$ layers, $n_{\text{heads}} = 12$ heads, embedding dimension $d_{\text{model}} = 768$). Both models are initialized using Mitchell initialization, do not use biases, and use a learnable class token and a patch size of 2.

**Training:** We train these models with a batch size of $B = 128$ for $100,000$ steps with optimization hyperparamters: $\beta_1 = 0.9$, $\beta_2 = 0.999$, $\epsilon = 10^{-8}$ and weight decay strength $\lambda = 0.01$. The learning rate is linearly increased from zero to a target learning rate $\eta$ in $T_{\text{wrm}} = 2048$ steps, followed by cosine decay to $\eta_{\text{min}} = \eta/10.0$.

## D.5 ESTIMATED COMPUTATIONAL RESOURCES

Each experiment required approximately 12 H100 GPU hours to complete. Our experimental design included around 8 learning rate variations, 2 distinct datasets for the four training tasks, resulting in 64 total runs. This amounted to 768 H100 GPU hours for the primary experiments. Additional small-scale exploratory experiments consumed approximately 250 H100 GPU hours, bringing the total computational resources used in this study to around 1000 H100 GPU hours.

# E THE *SlimAdam* OPTIMIZER

## E.1 *SlimAdam* ALGORITHM

This section describes the *SlimAdam* algorithm in detail. *SlimAdam* implementation consists of three steps. The code is available at https://github.com/ml-conf-authors/low-memory-adam.

**Step 1: Collect SNR statistics using a small proxy model**

First, we collect layer-wise SNR statistics using a small proxy model with a $10\times$ smaller learning rate than optimal. In theory, we would perform the SNR analysis at the optimal learning rate to determine compression rules, but this approach only saves around $30\%$ of seconds moments with a cutoff of 1.0 for Transformer models. Instead, we chose a $10\times$ smaller learning rate, which predicts saving around $99\%$ of second moments for a large range of cutoffs.

---

**Algorithm 1** Collect SNR statistics using a small proxy model

---

**Require:** Small model, dataset, optimization hparams ($10\times$ smaller learning rate)
1: Train for $T_{SNR}$ steps
2: **for all** layer $l$ in model **do**
3:     **for all** compression dimension $K \in \{(0,), (1,), (0,1)\}$ **do**
4:         Compute and Record $\text{SNR}_K(V_t^{(l)})$ according to Equation (2)
5:     **end for**
6: **end for**

---

**Step 2: Extract Compression Rules from SNR Statistics**

Next, we identify the compression dimension $K^*$ for each layer type with the highest SNR:

$$K^* = \arg\max_K \mathbb{E}_\tau[\text{SNR}_K(V^{(l)})]. \tag{7}$$

If SNR $\mathbb{E}_\tau[\text{SNR}_{K_{\max}}(V^{(l)})]$ exceeds the cutoff, we set the compression dimension $K^{(l)}$ to $K_{\max}$. Otherwise, no compression is performed. This results in consistent compression rules that generalize across depth and width and can be reused.

**Step 3: SlimAdam Optimizer**

Finally, we train the target model using Adam with shared second moments (Equation (1)) along these compression dimensions $K^*$. Given either (1) SNR-derived compression rules or (2) pre-computed rules from Table 1, SlimAdam applies these rules during training using Equation Equation (1). If $K^{(l)} = \varnothing$, SlimAdam does not compress second moments, and the optimization is identical to Adam.

For new training configurations, we suggest deriving compression rules using the SNR statistics of a smaller model. For known training setups, such as GPT pre-training, Table 1 rules can be used out of the box.

---

**Algorithm 2** Compression Rule Extraction from SNR Statistics.

---

**Require:** layer-wise SNR statistics and SNR cutoff
1: **for all** layer $l$ in model **do**
2:     $K^{(l)} \leftarrow \varnothing$
3:     **if** $\dim(V^{(l)}) > 1$ **then**
4:         $K_{\max} = \arg\max_K \mathbb{E}_\tau[\mathrm{SNR}_K(V_\tau^{(l)})]$
5:         **if** $\mathbb{E}_\tau[\mathrm{SNR}_{K_{\max}}(V^{(l)})] > \mathrm{cutoff}$ **then**
6:             $K^{(l)} \leftarrow K_{\max}$
7:         **end if**
8:     **end if**
9: **end for**
10: **return** $K^*$ for all layers

---

**Algorithm 3** SlimAdam

---

**Require:** Learning rate $\eta$, moment decay rates $\beta_1, \beta_2$, layer-wise compression rules $K^{(l)}$
1: **for** each training step $t$ **do**
2:     $G_t := \nabla_W L(\boldsymbol{\theta}_t)$
3:     **for** each layer $l$ **do**
4:         $M_{t+1}^{(l)} = \beta_1 M_t^{(l)} + (1 - \beta_1)G_t^{(l)}$
5:         **if** $K^{(l)} \neq \varnothing$ **then**
6:             $V_{t+1}^{(l)} = \beta_2 V_t^{(l)} + (1 - \beta_2)\mathbb{E}_{K^{(l)}}[(G_t^{(l)})^2]$
7:         **else**
8:             $V_{t+1}^{(l)} = \beta_2 V_t^{(l)} + (1 - \beta_2)(G_t^{(l)})^2$
9:         **end if**
10:         $\hat{M}_{t+1}^{(l)} \leftarrow \frac{M_{t+1}^l}{1 - \beta_1^t}$
11:         $\hat{V}_{t+1}^{(l)} \leftarrow \frac{V_{t+1}^{(l)}}{\sqrt{1 - \beta_2^t}}$
12:         $W_{t+1}^{(l)} = W_t^{(l)} - \eta_t \frac{\hat{M}^{(l)}{}_{t+1}}{\sqrt{\hat{V}_{t+1}^{(l)}} + \epsilon}$
13:     **end for**
14: **end for**

---

## E.2 EFFECT OF SNR CUTOFF AND PROXY MODEL LEARNING RATE ON *SlimAdam* PERFORMANCE

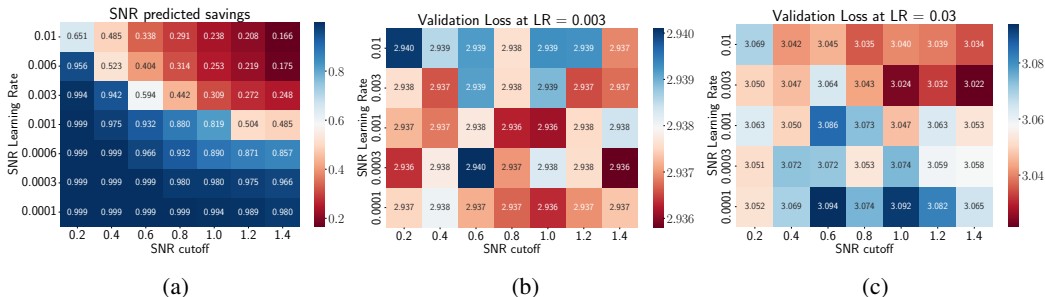

(a)        (b)        (c)

Figure 12: **Effect of SNR cutoff and proxy model learning rate on *SlimAdam* performance:** (a) SNR predicted memory savings, (b, c) validation loss as a function of SNR learning rate and cutoff for optimal learning rate and a large learning rate.

Figure 12 shows the effect of SNR cutoff and proxy model learning rate (SNR learning rate) on *SlimAdam* performance for GPT-small pre-trained on FineWeb-Edu.

## E.3 ADDITIONAL RESULTS FOR *SlimAdam*

This section provides additional results for Section 5. Figure 13 compares SNR predicted savings and performance of *SlimAdam* with other baselines on additional tasks. Figures 15 and 16 shows the training loss and downstream performance (HellaSwag and TruthfulQA) of Llama-3.2 1B and Llama 3.2 3B fine-tuned on the Alpaca dataset.

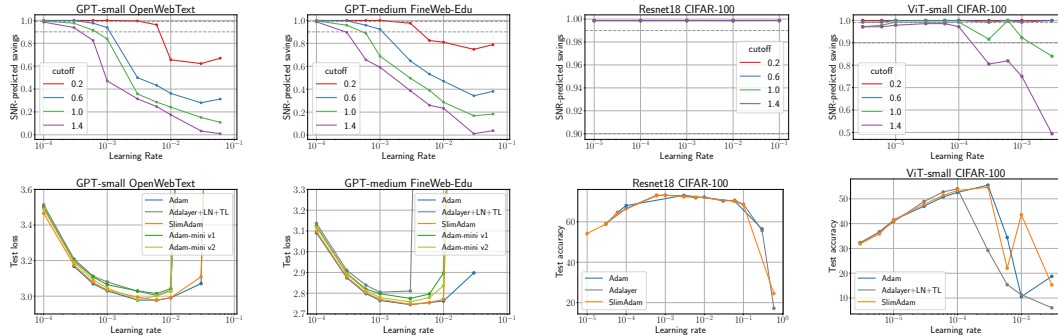

Figure 13: (Top) Fraction of second moments saved (relative to Adam) as a function of learning rate and SNR cutoff across training configuration, as suggested by the SNR analysis. (Bottom) Performance comparison across learning rates between SlimAdam and baselines.

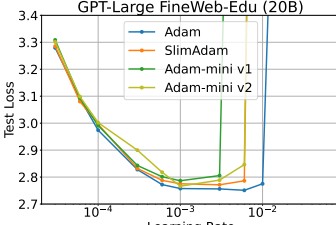

Figure 14: Performance comparison of Slimadam and baselines across learning rates for GPT-large model with 1B parameters trained on 20B tokens.

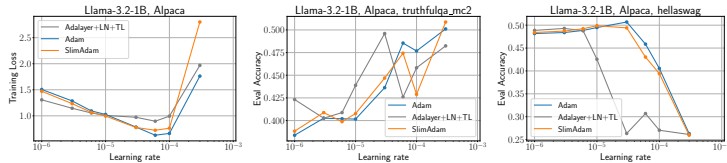

Figure 15: Training loss and Downstream performance of Llama-3.2 1B finetuned on the Alpaca dataset.

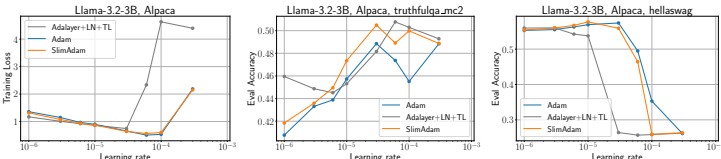

Figure 16: Training loss and Downstream performance of Llama-3.2 3B finetuned on the Alpaca dataset.

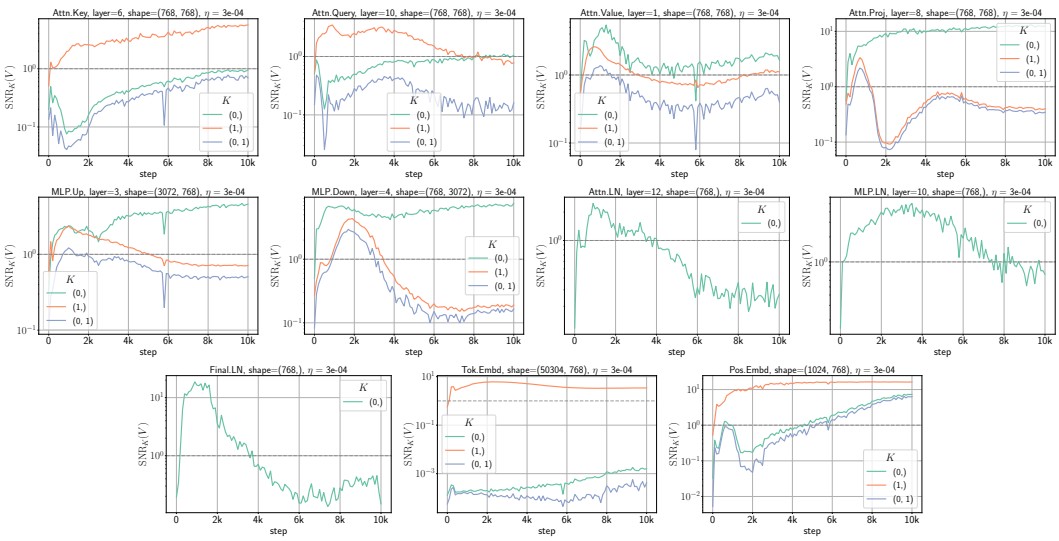

Figure 17: SNR trajectories of GPT-small trained on OpenWebText. For each layer type, the layer number is selected at random.

## F  SNR ANALYSIS OF DIVERSE TRAINING TASKS

### F.1  LANGUAGE PRE-TRAINING

This section provides supporting results for the SNR analysis of language pre-training performed in Section 3.1.1. We considered three experiments to explore the model size and dataset dependency on the SNR results:

1. GPT-small trained on OpenWebText (Figures 17 and 18)

2. GPT-small trained on FineWeb-Edu (Figures 19 and 20)

3. GPT-medium trained on FineWeb-Edu (Figure 21)

Figures 17 and 19 show that similar SNR trajectories are observed across different web text datasets. The layerwise trends shown in Figures 18 and 20 further support this claim. Furthermore, Figure 21 shows that similar SNR trends for a GPT-medium model.

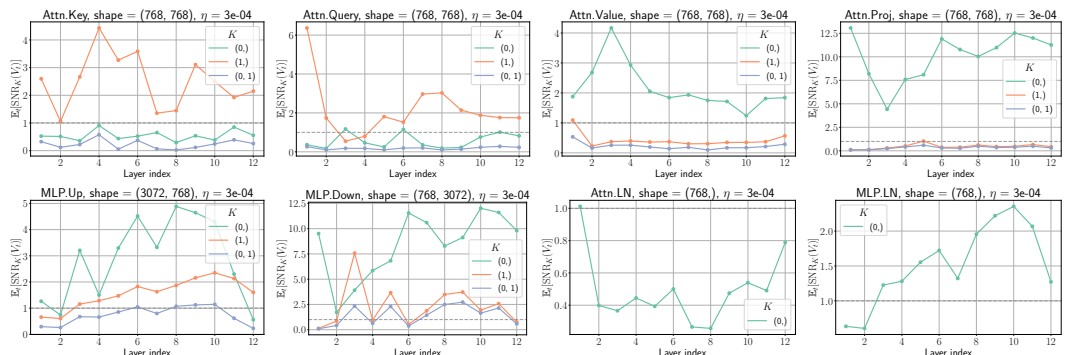

Figure 18: Layer dependence of averaged SNR values of GPT-small trained on OpenWebText.

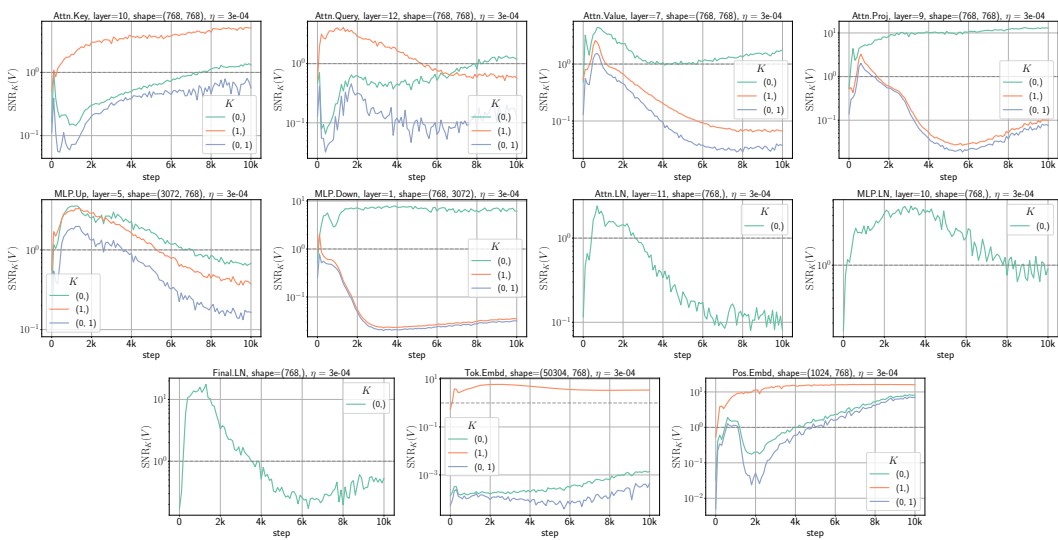

Figure 19: SNR trajectories of GPT-small trained on 10B subset of FineWeb-Edu. For each layer type, the layer number is selected at random.

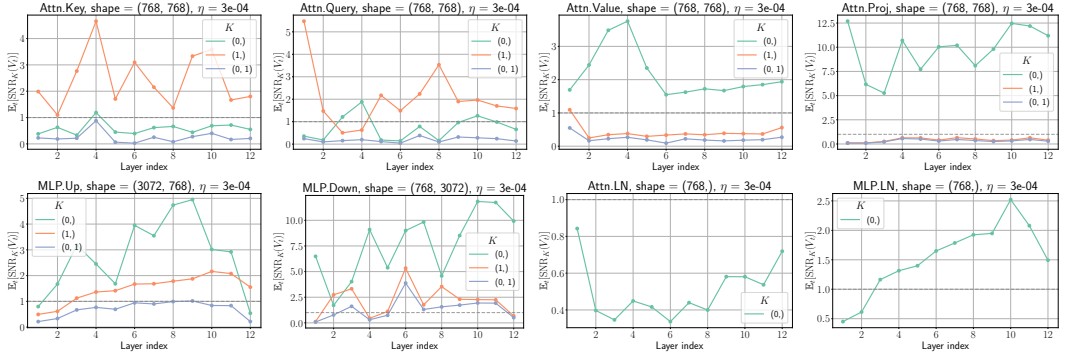

Figure 20: Layer dependence of averaged SNR values of GPT-small trained on 10B token subset of FineWeb-Edu.

## F.2 LANGUAGE FINE-TUNING

Figure 22 shows the SNR trends for pre-trained Llama 3.2 1B, fine-tuned on the Alpaca dataset. In comparison to the GPT pre-training experiments, we observe that the SNR values of attention key

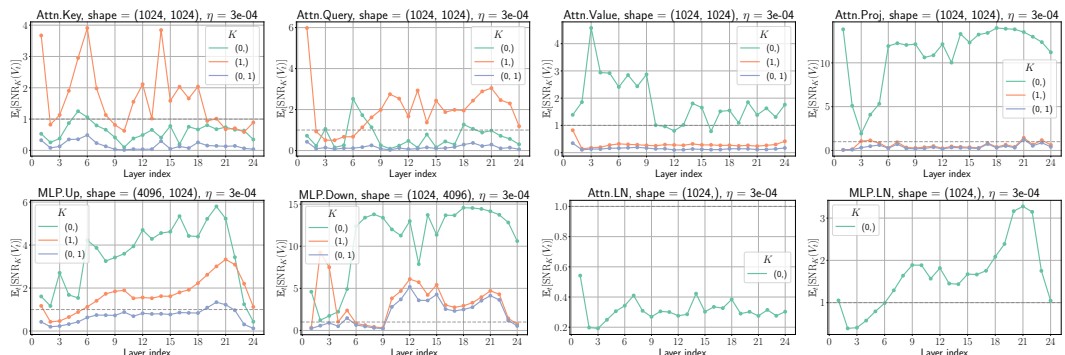

Figure 21: Layer dependence of average SNR values of the GPT-medium trained on FineWeb-Edu.

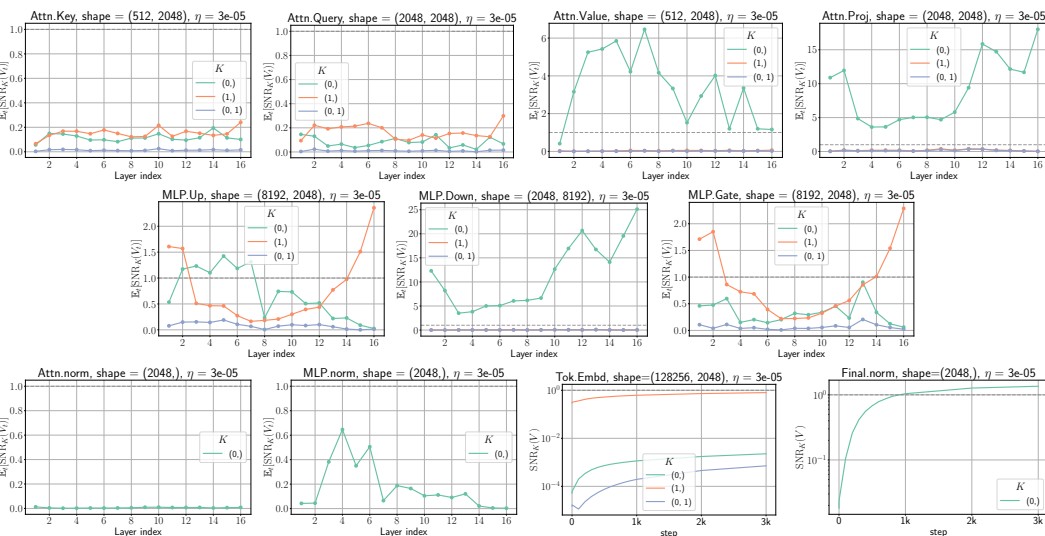

Figure 22: SNR analysis of pre-trained Llama 3.2 1B fine-tuned on Alpaca dataset.

and query second moments are significantly lower than 1.0. More generally, we observe lower SNR values, suggesting less compressibility.

## F.3   IMAGE CLASSIFICATION

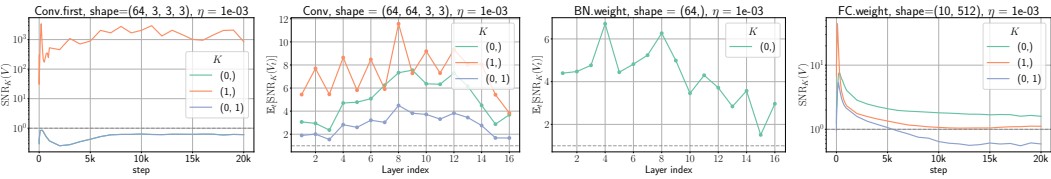

Figure 23: SNR trends of different layers of ResNet-18 trained on CIFAR-10.

Next, we examine the SNR trends of ResNets and ViTs trained on image classification tasks. As shown in Figures 23 and 24, ResNets trained on both CIFAR-10 and CIFAR-100 exhibit consistently high SNR values, suggesting compressibility. Most layers maintain high SNR values throughout training, with notable exceptions at the network boundaries. The first convolutional layer averses compressibility along the fan$_{out}$ dimension, while the final layer exhibits declining SNR values during later training stages when both dimensions are compressed. Unlike LayerNorm in Transformers, BatchNorm layers demonstrate SNR values around 1.0 throughout training.

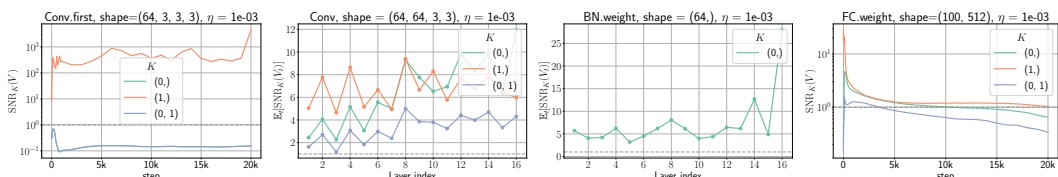

Figure 24: SNR trends of different layers of ResNet-18 trained on CIFAR-100.

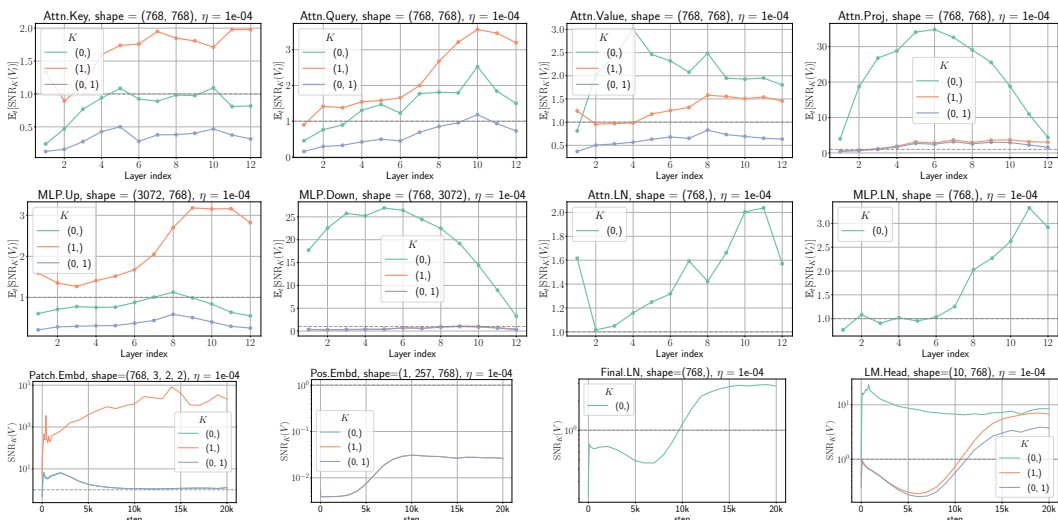

Figure 25: SNR trends of different layers of ViT-small trained on CIFAR-10.

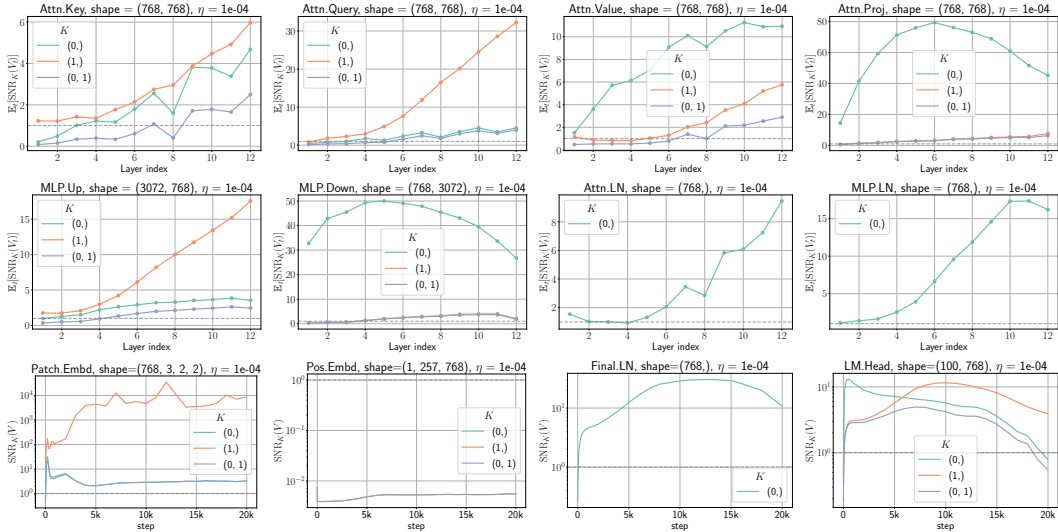

Figure 26: SNR trends of different layers of ViT-small trained on CIFAR-100.

# G EFFECT OF TRAINING HYPERPARAMETERS ON COMPRESSIBILITY

## G.1 LARGE LEARNING RATES REDUCE COMPRESSIBILITY

This section provides supporting results for Section 4.2 on the effect of learning rates on averaged SNR values $\mathbb{E}_t[\mathrm{SNR}_K(V_t)]$. For each layer, we analyze the effect of the learning rate on the dimension $K^*$ with the highest SNR. Figure 28 shows that the averaged SNR values consistently decrease with the learning rate. This decline suggests that higher learning rates cause training to explore regions

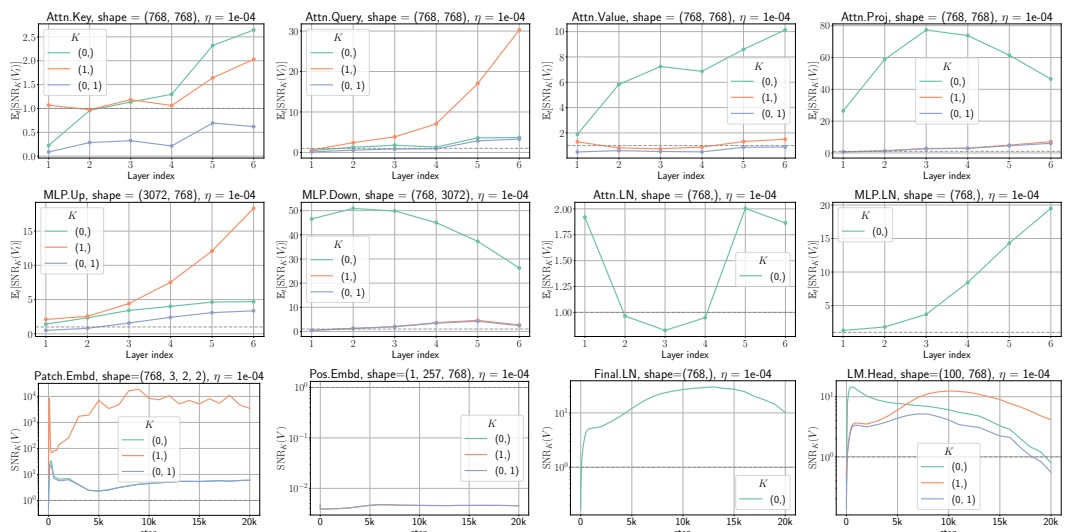

Figure 27: SNR trends of different layers of ViT-mini trained on CIFAR-100.

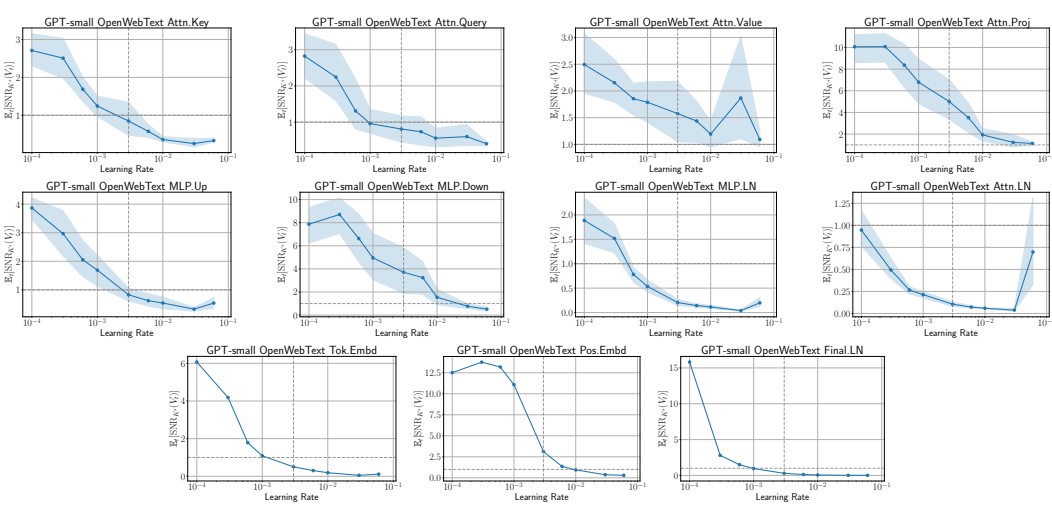

Figure 28: The effect of learning rate on the averaged SNR values of different layers of a GPT-small model trained on the OpenWebText dataset. For each layer, we have selected the dimension $K^*$ with the highest SNR. The shaded region around the mean trend shows the variation across depth. The vertical dashed line at 3e-03 denotes the optimal learning rate.

of parameter space where gradients contain more outliers, thereby reducing compression feasibility across all layers. Based on the effect of increasing the learning rate on SNR values, we classify layer types into two categories:

1. *Layers that exhibit low SNR values ($\lesssim$ 1) at the optimal learning rate:* Token Embedding/LM Head, LayerNorm, attention keys, queries and MLp.Up.

2. *Layers that exhibit high SNR values ($\gtrsim$ 1) even at the optimal learning rate:* Attention values, projections and MLP.Down.

## G.2 BATCH SIZE WITH OPTIMAL $\beta_2$ VALUE HAS A NOMINAL EFFECT ON COMPRESSIBILITY

In this section, we analyze the effect of batch size on SNR trends. We consider GPT models trained on 10B tokens of the FineWeb dataset for batch sizes $B \in \{32, 256, 1024\}$ and $\beta_2 \in \{0.95, 0.99, 0.999\}$. Consistent with prior work Porian et al. (2024), we observe that the optimal $\beta_2$ increases with decreasing batch size, as shown in Table 3. For a fair comparison, we use the optimal $\beta_2$ for each

batch size. Figure 29 shows that batch size has a nominal effect on the SNR trends for most layers. Table 3 also shows that SlimAdam matches Adam's performance while saving $99\%$ of second moments for batch sizes $B = 256$ and $1024$, while saving $92\%$ in the noisy regime of small batch size $B = 32$.

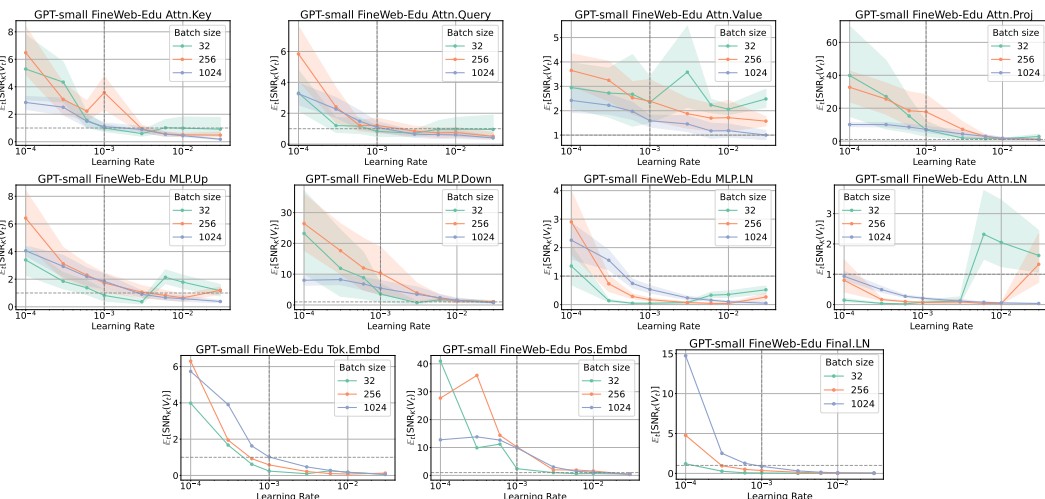

Figure 29: The effect of batch size on the averaged SNR values of different layers of a GPT-small model trained on the Fineweb dataset. For each layer, we have selected the dimension $K^*$ with the highest SNR. The shaded region around the mean trend shows the variation across depth. For each batch size, we select the optimal $\beta_2$ as described in Table 3.

## H  EFFECT OF INITIALIZATION ON COMPRESSIBILITY

This section provides supporting results for Section 4.3 on the effect of initialization on averaged SNR values $\mathbb{E}_t[\mathrm{SNR}_K(V_t)]$. We analyze how different initialization schemes affect SNR trends by comparing PyTorch's default initialization with the commonly used Mitchell initialization used in GPT models (recall that Mitchell initialization scales down the variance by $1/\mathrm{depth}$ in layers that add to the residual stream, such as Attn.Proj and MLP.Down). For simplicity, we select the dimension $K^*$ with the highest SNR for each layer.

Figure 30 shows that PyTorch's default initialization exhibits substantially lower SNR values across layers, especially the layers that add to the residual stream (Attn.Proj and MLP.Down) exhibit substantially lower SNR values. These results suggest that the compression feasibility depends on initialization choices and architectural details, suggesting that a single compression strategy is unlikely to work universally.

## I  TAILED TOKEN DISTRIBUTION REDUCE COMPRESSIBILITY

Figure 31 shows additional SNR trajectories for the token distribution experiment discussed in Section 4.1. For both layers, the SNR values along the token dimension ($K = 0$ for Tok.Embd and $K = 1$ for LM.Head) decrease as the vocabulary size is increased. This suggests that at large vocabulary sizes, each token evolves at its own pace and this requires its own effective learning rate.

Table 3: Performance comparison across different batch sizes for GPT-small trained on FineWeb.

| Batch Size | Optimal $\beta_2$ | Adam Loss | SlimAdam Loss | Second Moment Savings (%) |
|---|---|---|---|---|
| 32 | 0.999 | 2.961 | 2.960 | 92.8 |
| 256 | 0.99 | 2.956 | 2.958 | 99.9 |
| 1024 | 0.95 | 2.959 | 2.960 | 99.4 |

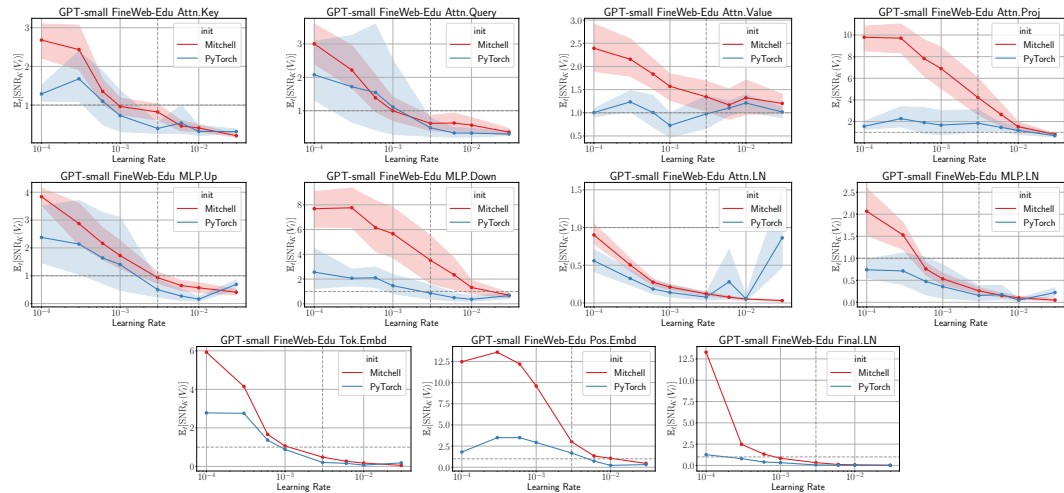

Figure 30: The effect of initialization on the averaged SNR values of different layers of a GPT-small model trained on the OpenWebText dataset. For each layer, we have selected the dimension $K^*$ with the highest SNR. The shaded region around the mean trend shows the variation across depth. The vertical dashed line at 3e-03 denotes the optimal learning rate for Mitchel initialization.

## J  ROBUSTNESS OF *SlimAdam* COMPRESSION RULES

This section analyzes the robustness of *SlimAdam* rules across datasets and model sizes. These variations disappear when using the depth-averaged SNR.

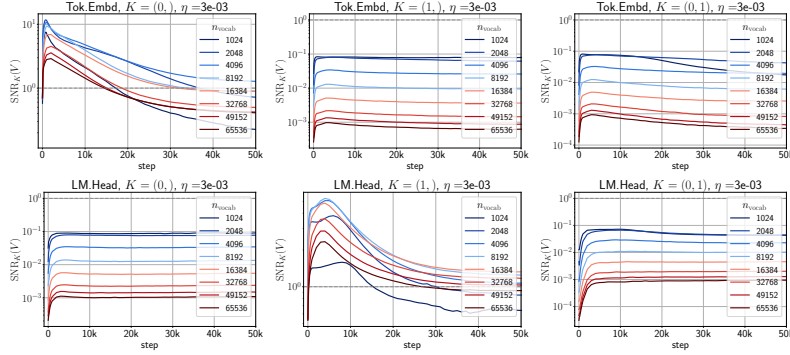

Figure 31: SNR trajectories of the token embedding and linear head of the simplified two-layer model with varying vocabulary sizes.

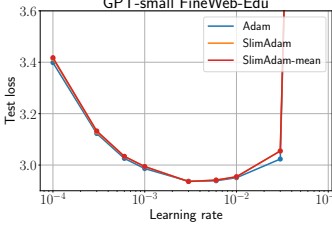

Figure 32: *SlimAdam* with compression rules derived from depth-averaged SNR per layer type (*SlimAdam-mean*) achieves identical performance to SlimAdam with per-layer compression rules.

## J.1 DATASET DEPENDENCY OF SLIMADAM RULES

This section analyzes how *SlimAdam*'s compression rules vary across different datasets. We compare rules derived from OpenWebText against FineWeb-Edu using GPT-small. The compression rules remain largely consistent, with differences in only five matrices, primarily in early MLP layers, as summarized in Table 4.

Table 4: Compression rule differences between datasets for GPT-small.

| Layer | OpenWebText | FineWeb-Edu |
|---|---|---|
| *Attention* | | |
| Attn Query (L3) | None | fan-out |
| *MLP* | | |
| MLP Up (L0) | fan-out | None |
| MLP Up (L1) | None | fan-out |
| MLP Proj (L1) | fan-out | fan-in |
| MLP Proj (L2) | fan-in | fan-out |

## J.2 WIDTH DEPENDENCY OF SLIMADAM RULES

This section analyzes the robustness of *SlimAdam*'s compression rules across model widths ($d_{\text{model}}$). We compare the SNR-derived compression rules for GPT-small with embedding dimension $d_{\text{model}} = 768$ against a narrower model ($d_{\text{model}} = 256$. Out of all layer matrices, we observe differences in compression rules for only 12 matrices, primarily in early to middle layers, as shown in Table 5.

Table 5: *SlimAdam* compression rule differences between narrow (width 256) and wide (width 768) models.

| Layer | $d_{\text{model}} = 256$ | $d_{\text{model}} = 768$ |
|---|---|---|
| *Attention Components* | | |
| Attention Value (L0) | fan-in | fan-out |
| Attention Key (L2) | fan-out | fan-in |
| Attention Query (L2) | fan-in | fan-out |
| Attention Query (L3) | fan-in | None |
| *MLP Components* | | |
| MLP Up (L0) | fan-in | fan-out |
| MLP Up (L1) | fan-out | None |
| MLP Proj (L2) | fan-out | fan-in |
| MLP Up (L3) | fan-in | fan-out |
| MLP Up (L4) | fan-in | fan-out |
| MLP Proj (L4) | fan-in | fan-out |
| MLP Proj (L5) | fan-in | fan-out |
| MLP Up (L6) | fan-in | fan-out |

The variations observed in Tables 4 and 5 can be eliminated by deriving compression rules using depth-averaged SNR for each layer type. Figure 32 shows that compression rules derived from depth-averaged SNR result in identical performance to SlimAdam with per-layer compression rules.

## K WEIGHT-SPACE DISTANCE BETWEEN SLIMADAM AND ADAM

Figure 33 shows that the normalized weight space distance between SlimAdam and Adam increases during training, suggesting that they learn different solutions. We leave a comprehensive study on the analysis and implication of this observation for future work.

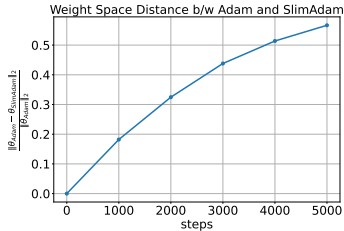

Figure 33: Normalized weight space distance between Adam and SlimAdam increases with training.

## L TRAINING TRAJECTORIES OF SLIMADAM

Recommended version: This section presents the training trajectories of loss and accuracy corresponding to the results in Figure 9. Figures 34, 35, 36, and 37 show these trajectories for GPT pre-training, LLaMA finetuning, ResNet, and ViT image classification, respectively.

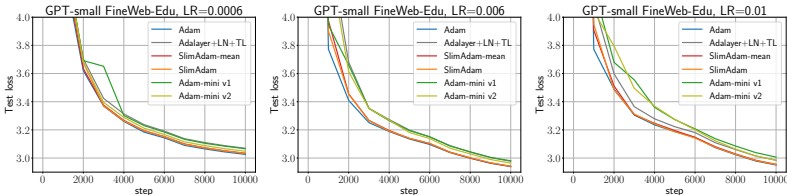

Figure 34: Loss trajectories for the GPT pre-training task corresponding to the results in Figure 9.

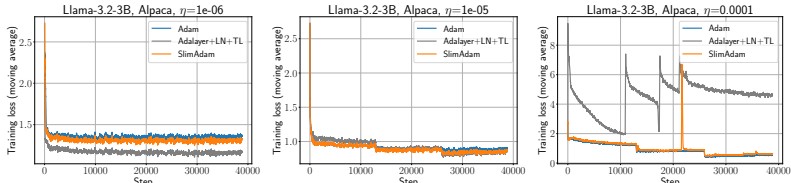

Figure 35: Smoothed loss trajectories (last 100 steps) for the Llama finetuning task corresponding to the results in Figure 9.

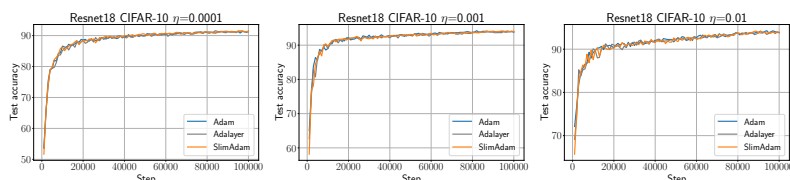

Figure 36: Accuracy trajectories for the ResNet classification task corresponding to the results in Figure 9.

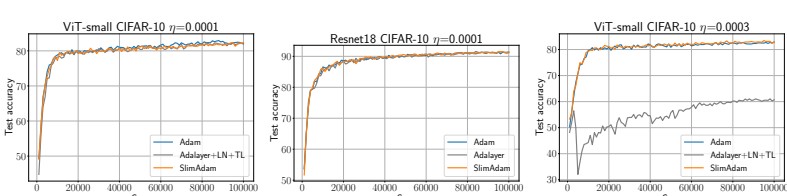

Figure 37: Accuracy trajectories for the ViT classification task corresponding to the results in Figure 9.

