# OpenReview forum: "When Can You Get Away with Low Memory Adam?"
_ICLR.cc/2026/Conference — Submitted to ICLR 2026_

### Official Review · Reviewer_XBFB · 2025-10-15

**Soundness:** 3
**Presentation:** 2
**Contribution:** 2
**Rating:** 2
**Confidence:** 4

**Summary:**

This paper proposes a layer-wise Signal-to-Noise Ratio (SNR) analysis to determine when the second-moment tensors in optimization algorithms (e.g., Adam) can be compressed by replacing them with their dimensional means. Given that SNR is computed as $\text{mean} / \text{variance}$, it serves as a natural metric for this purpose: a high SNR indicates that a tensor can be effectively approximated by its mean without significant performance loss. This approach provides a practical, quantitative guide for compressing optimizer states and offers evidence that Adam may not always require full second-moment information.

**Strengths:**

This work proposes to apply SNR as a metric to guide the compression of second-moment tensors with their means in LLM training. It offers a threshold-based criteria to determine when and how such mean compression can be applied across different architectural components of LLMs (e.g., query, key, value, and MLP layers). Furthermore, this work empircially investigates several factors influencing compressibility, including learning rate, data distribution, and initialization.

**Weaknesses:**

1. The main motivation of this work is to establish a metric for guiding dimension-wise mean compression of second-moment tensors and provide SlimAdam. However, this goal appears to overlap with Adam-mini [1], which not only implemenst a compression method based on block-wise mean values but also provides insights based on Hessian structure to explain why the full second-moment may be unnecessary and to guide how to compress. The authors should more clearly delineate their contributions and explicitly contrast their approach with the insights and methods provided by Adam-mini.

2. SNR is a natural choice for quantifying the viability of mean compression, given its formula of $\text{mean} / \text{variance}$. The paper does not sufficiently justify why it is superior to other plausible metrics. For instance, measures based on the L2-norm or KL-divergence of the error introduced by compression, or just variance, could be more direct and computationally efficient. The author should demonstrate the unique advantages of SNR over other alternatives through theoretical analysis or empirical comparision.

3. For mean compression, it is clear that higher SNR correlates with better compressibility. However, the method remains dependent on an empirically set threshold (e.g., $\alpha=1$) to make compression decisions. This dependency not only limits the generality of the method by introducing a potentially sensitive hyperparameter across different scenarios but also raises the question of whether other metrics (e.g., L2-norm, KL-divergence, or variance) could perform just as effectively with a similarly tuned threshold.

4. The utility of SNR seems limited to mean compression and may not extend to or guide other compression paradigms (e.g., low-rank factorization, quantization).

[1] Zhang, Yushun, et al. "Adam-mini: Use fewer learning rates to gain more." arXiv preprint arXiv:2406.16793 (2024).

**Questions:**

1. The choice of SNR is intuitive for mean replacement, but why is it superior to other direct measures of compression error, such as the L2-norm or KL-divergence between the original and compressed tensor? Could the authors provide either (a) an empirical ablation study comparing the compression guidance performance of SNR against these other metrics, or (b) a theoretical argument for why SNR is an optimal or more robust criterion?

2. If the performance of the method is similar when using a simple threshold on other metrics (e.g., compress if $\text{variance} < X$, compress if $\text{L2-norm of error introduced by compression}/\text{L2-norm of target tensor}$), does this suggest the core insight is about identifying low-variance parameters rather than the unique information provided by SNR? What is the specific advantage of the SNR ratio over just using the variance or standard deviation?

3. The threshold $\alpha=1$ is presented as a critical value for making compression decisions. How was this value determined? Is it robust across different model architectures, layers, and tasks? Could the authors show sensitivity analyses for this threshold to demonstrate its generality?

4. The presentation could be improved for better clarity and reproducibility. For example, using pseudocode rather than plain text would help readers better understand the algorithm's workflow.

---

> ### Author Response · Authors · 2025-11-21
>
> We thank the reviewer for their time in reviewing our work and providing feedback. We appreciate their acknowledgement that our proposed metric is natural and that our work serves as a practical, and quantitatively grounded guide for identifying memory savings when using Adam. The concerns they raise are thoughtful and so we want to address each one individually below.
>
> > The goal appears to overlap with Adam-mini .... The authors should more clearly delineate their contributions and explicitly contrast their approach with the insights and methods provided by Adam-mini.
>
> While Adam mini also performs dimension-wise mean compression of second moments, our work complements and extends Adam-mini in several important ways:
> 1. **Different approach**: Adam-mini uses Hessian spectrum analysis at small scales, then transfers block-wise compression rules to larger models. In contrast, our SNR analysis is computationally cheaper than Hessian computation and can be applied directly to billion-parameter models (Figure 13) with minimal computational and memory overhead. This makes our approach more accessible for practitioners working with large models where Hessian computation is prohibitive.
> 2. **Better compression rules lead to superior stability**: Our SNR-derived rules differ from Adam-mini for a few layers (Table 1): we compress attention values/projections along fan-out vs Adam-mini's fan-in, and we leave LayerNorms uncompressed vs Adam-mini's default compression. Figure 1(c) and Figure 8 demonstrate that this matters at Adam’s optimal learning rate. SlimAdam maintains Adam-level stability at large learning rates where Adam-mini exhibits training instabilities.
> 3. **Dynamic view of compressibility**: A key contribution of our work is that Adam’s compressibility changes dramatically with hyperparameters (vocabulary size, learning rate and initialization) and training tasks. Learning rate has the largest effect on Adam’s compressibility. The overall compressibility reduces to ~30% (Figure 8 top left) at large learning rates from ~99% observed at small learning rates. Furthermore, we also show that compressibility reduces at large vocabulary sizes and suboptimal initializations. Overall, this gives practical guidance for practitioners: when using low memory optimizers, use good initializations and train at smaller learning rates.
> 4. **Diverse evaluation across tasks**: We systematically analyze four training domains (GPT pre-training, Llama fine-tuning, ResNet, ViT), revealing that optimal compression strategies are task-dependent (Table 1). Adam-mini focuses primarily on language modeling. We also take care to sanity check our SNR measure by confirming that it predicts well-known results like the fact that the simpler optimization landscape of ResNets needs minimal preconditioning (fully compressible moments, can use SGD).
>
> > The presentation could be improved for better clarity and reproducibility. For example, using pseudocode rather than plain text would help readers better understand the algorithm's workflow.
>
> We thank the reviewer for this suggestion. The complete code is publicly available at https://github.com/ml-conf-authors/low-memory-adam. We note that SlimAdam differs from standard Adam by only one line: replacing the second moment update (Equation 1) with dimension-wise averaging when appropriate. If the reviewer believes converting the Appendix algorithms to a specific pseudocode format would significantly improve clarity, we are happy to do so. Furthermore, we have significantly improved the description of SlimAdam's three-step process in Section 5 and added a flowchart visualizing the complete SlimAdam method, including when to derive new rules versus reusing existing ones.

---

> ### Author Response · Authors · 2025-11-21
>
> > SNR over simple variance, L2, and KL
>
> > The choice of SNR is intuitive for mean replacement, but why is it superior to other direct measures of compression error, such as the L2-norm or KL-divergence between the original and compressed tensor? Could the authors provide either (a) an empirical ablation study comparing the compression guidance performance of SNR against these other metrics, or (b) a theoretical argument for why SNR is an optimal or more robust criterion?
>
> We thank you for your suggestion to use variance instead of SNR, but the variance is fundamentally unsuitable for our purpose due to dimensional scaling issues. The core problem is that the gradient magnitudes scale differently across layers and with model size. Ref. [1] shows that gradients in different layers exhibit different width-dependent (see their Table 1).  Consequently, second-moment variances inherit these scaling properties, making any fixed variance threshold arbitrary and non-transferable. Similar to the variance, the L2 norm and KL divergence also face the same dimensional scaling issues. By comparison, **SNR is a dimensionless quantity** and does not suffer from this issue.
>
> Let's consider a concrete example. Consider two layers: (1) layer A with gradients that have a variance of 1/width, and (2) layer B that has a gradient variance of 1. In this setting, the variance of the second moments scales as 1/width and 1 for the two layers. Using the variance has two critical issues:
> 1. A threshold that works for Layer A fails for Layer B
> 2. Even for a single-layer type, the threshold must be re-tuned as model width changes.
>
> By comparison, SNR remains independent of width and suggests a principled cutoff for compression: when SNR > 1, the mean term dominates the variance regardless of the absolute scale of the gradients. Our empirical results confirm this theoretical insight: SNR = 1.0 works universally across all architectures (GPT, Llama, ViT, ResNet) from 100M to 1B parameters, without any tuning.
>
> Furthermore, we now show that the relative L2 compression error is equal to 1/(1+SNR) in Section 3 and Appendix C.1. This provides a theoretical justification for using SNR as a measure of compression -- higher SNR results in lower relative compression error, whereas lower SNR suggests significant information is lost during compression.
>
> [1] Scaling Exponents Across Parameterizations and Optimizers, ICML 2024
>
> > For mean compression, it is clear that higher SNR correlates with better compressibility. However, the method remains dependent on an empirically set threshold (e.g., $\alpha=1$) to make compression decisions. This dependency not only limits the generality of the method by introducing a potentially sensitive hyperparameter across different scenarios but also raises the question of whether other metrics (e.g., L2-norm, KL-divergence, or variance) could perform just as effectively with a similarly tuned threshold.
> > The threshold $\alpha=1$ is presented as a critical value for making compression decisions. How was this value determined? Is it robust across different model architectures, layers, and tasks? Could the authors show sensitivity analyses for this threshold to demonstrate its generality?
>
> We thank the reviewer for their question. We would like to emphasize that the cutoff $\alpha = 1$ is a principled choice with a theoretical justification: SNR > 1 indicates that the mean term dominates the variance, suggesting that compression is possible. Furthermore, SNR is a dimensionless quantity, and therefore, we do not expect it to change appreciably with the scale of the gradients. Importantly, we show that the SNR = 1 cutoff works out of the box, without any manual tuning (Figure 8).
>
> In contrast, L2-norm, KL-divergence, or variance would require setting thresholds separately for each layer type due to dimensional scaling. Even if both approaches require "one hyperparameter" per layer, the key difference is tuning complexity: SNR requires tuning a single global cutoff (as its invariant of the second moment scale), whereas dimension-dependent metrics would require layer-wise cutoffs. This makes SNR far more practical and generalizable.

---

> > ### Comment · Reviewer_XBFB · 2025-11-24
> >
> > Thanks for your reply. I appreciate the clarification on those points. To make sure we're fully aligned, I'm following up on my previous questions:
> >
> > - About Adam-mini:
> >
> > In my understanding, Adam-mini proposes using the block-wise mean of the second-moment estimate to compress it during training. To support this approach, the authors empirically investigate the structure of Hessian matrices, finding that they are often approximately block-diagonal. This structure justifies the use of a block-wise mean as an effective approximation for a Newton update step. Although inspired by the structure of the Hessian, Adam-mini is designed to function without explicitly evaluating it.
> >
> > - About other metrics
> >
> > It is excellent that you provide a theoretical relationship between the SNR and the relative approximation error under the L2-norm (noted as Relative-Error-L2). However, this relationship directly raises a question about the necessity of using SNR. As you stated, Relative-Error-L2 = 1 / (1+SNR). Your insight is that the mean approximation can be used when SNR > 1, which is equivalent to requiring that the Relative-Error-L2 be less than 0.5. Furthermore, Relative Error-L2 is a straightforward and widely used guiding metric in adaptive approximation optimization, as seen in [1].
> >
> > Given these two concerns, I am unable to proceed with updating the score at this time.
> >
> >
> > [1] Refael, Yehonathan, et al. "Adarankgrad: Adaptive gradient-rank and moments for memory-efficient llms training and fine-tuning." arXiv preprint arXiv:2410.17881 (2024).

---

> > > ### Author Response · Authors · 2025-12-01
> > >
> > > We thank the reviewer for their comments and continued engagement. Below are our responses to the followup comments:
> > >
> > > > In my understanding, Adam-mini proposes using the block-wise mean of the second-moment estimate to compress it during training. To support this approach, the authors empirically investigate the structure of Hessian matrices, finding that they are often approximately block-diagonal. This structure justifies the use of a block-wise mean as an effective approximation for a Newton update step. Although inspired by the structure of the Hessian, Adam-mini is designed to function without explicitly evaluating it.
> > >
> > > We appreciate your summary of Adam-mini's approach. Our response above highlights how our work complements and contrasts with Adam-mini:
> > > - A more scalable analysis method applicable to billion-parameter models (Section 3)
> > > - Improved compression rules leading to better stability at large learning rates (Section 5)
> > > - Systematic analysis of how compressibility depends on hyperparameters and training tasks (Section 4)
> > > - Broader analysis across diverse domains beyond language modeling (Section 3)
> > >
> > >
> > > > It is excellent that you provide a theoretical relationship between the SNR and the relative approximation error under the L2-norm (noted as Relative-Error-L2). However, this relationship directly raises a question about the necessity of using SNR. As you stated, Relative-Error-L2 = 1 / (1+SNR). Your insight is that the mean approximation can be used when SNR > 1, which is equivalent to requiring that the Relative-Error-L2 be less than 0.5. Furthermore, Relative Error-L2 is a straightforward and widely used guiding metric in adaptive approximation optimization, as seen in [1].
> > >
> > >
> > > We thank the reviewer for sharing the related work. While the referenced work also utilizes relative error for compression, there are important distinctions:
> > >
> > > 1. Different compression schemes: The referenced work performs low-rank approximation via SVD, while SlimAdam does dimension-wise averaging. SNR requires only mean and variance calculations, whereas Adarankgrad's approach requires SVD decomposition at each step, which is computationally expensive. These are fundamentally different operations requiring different error metrics.
> > >
> > > 2. Different goals and scope: Our primary goal is to understand when and why Adam’s second moments can be compressed. Our SNR analysis reveals that second-moment compressibility varies systematically across training tasks, architectures (Section 3), and hyperparameters (Section 4), providing fundamental insights into Adam's behavior that are not addressed by the referenced work.

---

### Official Review · Reviewer_BcCm · 2025-10-27

**Soundness:** 3
**Presentation:** 2
**Contribution:** 2
**Rating:** 4
**Confidence:** 3

**Summary:**

This paper studies the signal-to-noise ratio of Adam’s second-moment tensors for every layer along the input-channel (column) and output-channel (row) directions, finding instances where entries exhibit low variance relative to their mean and can safely share statistics during training; using these SNR profiles, it introduces SlimAdam, which collapses second moments only on the high-SNR direction of each layer, cutting memory while preserving Adam-level convergence and accuracy.

**Strengths:**

The paper tackles a well-motivated problem—the large memory footprint of Adam’s second-moment matrices—and clearly pinpoints instances where collapsing each layer’s parameters to a single scalar shrinks an matrix to just or values, cutting memory use while retaining Adam-level accuracy and stability.

**Weaknesses:**

Since there are almost no changes from my last review of the paper, I'll keep the core of my argument. I’ll split my review into two parts — one on the empirical analysis and one on the proposed optimizer.

### Empirical analysis and design rationale

The paper’s primary findings are almost entirely empirical, and this lack of theory leaves several key decisions unclear—especially because the empirical signals themselves are not particularly strong. Axis sharing is constrained to whole fan-in or fan-out dimensions purely for implementation convenience, with no exploration of alternative groupings or proof that these axes are optimal (e.g., would results change if one considers randomly partitioning a layer’s parameters into two equal-size groups?). Similarly, the paper adopts (the interesting metric) SNR with a heuristic threshold as the only compression criterion, although simple variance (which answers “How much l2 loss do we pay if we collapse this vector to a scalar?”) would align more directly with the intuition the authors cite (“If entries along a dimension exhibit low variance relative to their mean, they can be effectively represented by a single value”).

Learning rate is the only hyperparameter the paper systematically analyzes. It is presented as the dominant knob that shifts SNR and thus determines which layers can be compressed, yet the text provides no a-priori reason why learning rate—rather than, say, Adam’s momentum coefficients or the batch size—should hold that position.


### Optimizer details
The optimizer requires selecting a compression axis for each layer or layer type. Choosing a compression axis means either relying on proxies or heuristics, or collecting fresh SNR statistics. My concern is that the latter defeats the purpose in some cases, and the former is not reliable.

Alternatively, we can train a small proxy model or reuse generic rules. Yet the authors themselves show that preferred compression axes shift with dataset, width, and vocabulary size. Even within the same dataset and width, layers of the same type show different preferences. Depth-averaging does not fully solve the problem for users operating at the tightest memory margins or in domains whose depth-specific SNR patterns have not been studied yet. Even the stronger patterns they find—for example, compressing along the embedding dimension versus the token dimension—may not yield an SNR above the cutoff needed to justify compression.

Full-size SNR collection defeats the purpose. To decide the sharing axis, you must first run the uncompressed model under standard Adam long enough to gather per-layer SNR statistics. During this warm-up, you still store the full second-moment tensors, so the memory spike SlimAdam tries to avoid is paid up front. For practitioners who want to fit a slightly larger model into fixed hardware, this spike means the maximum model size is still bounded by Adam’s footprint during the warm-up, undermining the value of a lighter optimizer.

**Questions:**

Please address my concerns above, especially around the empirical nature of the evidence, axis selection, and practicality at tight memory budgets. In addition, a high-level clarification would help: how should we interpret “compressibility” in this work beyond plots of SNR? In other words, is SNR a sufficient observable for when per-parameter adaptivity is redundant, and how does its dependence on learning rate versus other hyperparameters shape the generality of your claims?

---

> ### Author Response · Authors · 2025-11-21
>
> We thank the reviewer for their continued engagement with our work. However, we must respectfully note that we do not believe that the comment: "there are almost no changes from my last review” reflects the actual level of revision we have performed and specifically ignores the additional hyperparameter sensitivity experiments added since the draft was last reviewed. We will discuss these points in detail below.
>
> > Learning rate is the only hyperparameter the paper systematically analyzes. It is presented as the dominant knob that shifts SNR and thus determines which layers can be compressed, yet the text provides no a-priori reason why learning rate—rather than, say, Adam’s momentum coefficients or the batch size—should hold that position.
>
>  In addition to learning rate, we systematically analyzed the effect of batch size and $\beta_2$ in Appendix G.2, showing that SNR trends do not vary appreciably with batch size and learning rate remaining the primary factor reducing compressibility (summarized in Section 4.2). We summarize the experimental results below for completeness.
>
> We considered GPT models trained on the FineWebEdu dataset with batch sizes $B \in [32, 256, 1024]$ and $\beta_2 \in [0.95, 0.99, 0.999]$. For each batch size, we find the optimal $\beta_2$. Figure 29 shows that the batch size has a nominal effect on SNR trends. We also analyzed SlimAdam’s performance across these hyperparameters in Table 3, showing that SlimAdam matches Adam’s performance while saving 99% of second moments for batch sizes B = 256 and 1024, while saving 92% in the noisy regime of small batch size $B = 32$.
>
> > SNR over simple variance
>
> We thank you for your suggestion to use variance instead of SNR, but the variance is fundamentally unsuitable for our purpose due to dimensional scaling issues. The core problem is that the gradient magnitudes scale differently across layers and with model size. Ref. [1] shows that gradients in different layers exhibit different width-dependent (see their Table 1).  Consequently, second-moment variances inherit these scaling properties, making any fixed variance threshold arbitrary and non-transferable. By comparison, **SNR is a dimensionless quantity and does not suffer from this issue**.
>
> Let's consider a concrete example. Consider two layers: (1) layer A with gradients that have a variance of 1/width, and (2) layer B that has a gradient variance of 1. In this setting, the variance of the second moments scales as 1/width^2 and 1 for the two layers. Using the variance has two critical issues:
>
> 1. A threshold that works for Layer A fails for Layer B
> 2. Even for a single-layer type, the threshold must be re-tuned as model width changes.
>
> By comparison, SNR remains independent of width and suggests a principled cutoff for compression: when SNR > 1, the mean term dominates the variance regardless of the absolute scale of the gradients. Our empirical results confirm this theoretical insight: SNR = 1.0 works universally across all architectures (GPT, Llama, ViT, ResNet) from 100M to 1B parameters, without any tuning.
>
> Furthermore, we now show that the relative L2 compression error is equal to 1/(1+SNR) in Section 3 and Appendix C.1. This provides a theoretical justification for using SNR as a measure of compression -- higher SNR results in lower relative compression error, whereas lower SNR suggests significant information is lost during compression.
>
> [1] Scaling Exponents Across Parameterizations and Optimizers, ICML 2024
>
> > The authors themselves show that preferred compression axes shift with dataset, width, and vocabulary size. Even within the same dataset and width, layers of the same type show different preferences.
>
> We would like to take the opportunity to clarify that the preferred compression trends remain consistent for a given training task, but they vary across domains. Furthermore, while we show that the overall compressibility reduces with learning rate, vocabulary size, and suboptimal initialization, the preferred compression dimensions remain the same for a given training task. For example, Figure 31 shows that the embedding dimension remains as the preferred compression dimension as the vocabulary size increases, while the overall compressibility reduces.
>
> **Consistent trends for a training task**:
> As we discussed in Section 5, on varying the dataset and model size, we observe minor differences in preferred compression dimensions for a few layers. These variations disappear when using the depth-averaged SNR, matching those in Table 1 (further details in Appendix J).
>
> **Domain-specific patterns reflect optimization landscape differences:** Compressibility patterns varying across domains (pre-training vs fine-tuning vs image classification) are expected, as they reflect differences in optimization landscapes. Our cross-domain analysis prevents overfitting to any single setting and provides practitioners with intuition about what to expect when switching domains.

---

> > ### Comment · Reviewer_BcCm · 2025-11-22
> >
> > >We thank the reviewer for their continued engagement with our work. However, we must respectfully note that we do not believe that the comment: "there are almost no changes from my last review” reflects the actual level of revision we have performed and specifically ignores the additional hyperparameter sensitivity experiments added since the draft was last reviewed. We will discuss these points in detail below.
> >
> >
> > Before I respond to your comments, could you point me to the specific places in the main text where you made changes? If there were any major changes that I missed in the main text, could you flag those as well?

---

> ### Author Response · Authors · 2025-11-21
>
> > On the "full-size SNR collection defeats the purpose" concern
>
> The reviewer appears to have missed our discussion of a core component of the full methodology. **We explicitly specify using small proxy models for SNR collection (Step 1, Section 5) and not full-size models**. This design is motivated by significant empirical evidence that rules generalize from smaller to larger models for a given training task (e.g., language pre-training), while variations are observed across training tasks. This avoids the memory spike concern entirely, as practitioners never need to run the full uncompressed model.  The memory cost of Step 1 is negligible compared to training the target model. We have significantly improved the description of SlimAdam's three-step process (SNR collection, rule extraction, deployment) and clarified how practitioners can reuse existing rules (Table 1) for known tasks. We have also added a flowchart visualizing the complete SlimAdam method, including when to derive new rules versus reusing existing ones.
>
> > Axis selection /  Fan-in/fan-out design rationale
>
> **Our fan-in/fan-out design is not only practical from an implementation perspective but also enables architectural interpretability**, as it aligns with the activation and gradient flow through the network. For instance, it led to the discovery that each token in embedding-like layers requires its own second moment (equivalent to the embedding dimension, which is safe to compress), rather than leaving entire matrices uncompressed as suggested in prior work.
>
> > On the random partitioning baseline
>
> A random partitioning into two equal-sized groups would only reduce 50% ($N^2/2$) second moments, rather than our O(N)  reduction, making it an unfair comparison. A fair baseline would compare random groupings with equivalent memory usage, but we're uncertain what a fair comparison would be. We are open to alternative comparisons if there are any suggestions with an equivalent memory usage.
>
> > On interpreting compressibility beyond SNR
>
> We thank the reviewer for this question. In this work, we focus on a specific notion of compressibility: when second moments in a matrix can be replaced by their mean across fan-in or fan-out dimensions. While this is a constrained definition, it has a direct interpretation in terms of loss landscape geometry. High SNR indicates that matrix parameters along a dimension exhibit similar curvature, and it is safe to navigate these dimensions in the loss landscape with a shared effective learning. As we show in Appendix C.2, SNR relates to the condition number of Adam's preconditioner. High SNR means the preconditioner acts approximately like uniform scaling along that dimension, suggesting the local geometry is well-conditioned. In other words, parameters along that dimension learn at the same speed and don't need individual ‘learning rates’. Conversely, low SNR reveals when the landscape geometry is more complex and per-parameter adaptivity becomes necessary.
>
>
> > On whether SNR is sufficient
>
> Several results in our paper validate SNR as a meaningful measure:
>
> - **SNR relates to the relative compression error**: We now show that the relative L2 compression error is given by 1/(1+SNR) (see  Section 3 and Appendix C.1). This provides a theoretical justification for using SNR as a measure of compression -- higher SNR results in lower relative compression error, whereas lower SNR suggests significant information is lost during compression.
> - **ResNets provide a sanity check**: ResNets are known to have smooth optimization landscapes and can be trained effectively with SGD. Our SNR analysis reveals consistently high SNR values across ResNet layers, confirming that the metric correctly identifies cases where per-parameter adaptation is unnecessary.
> - **SlimAdam's empirical success**: SlimAdam matches Adam's performance while saving up to 99% of second moments across diverse tasks (Section 5). This demonstrates that SNR successfully identifies redundant adaptivity in practice: if SNR were an insufficient metric, we would observe performance degradation.
> - **Vocabulary size experiment validates SNR's predictive power**: We show that as vocabulary increases, SNR along the token dimension decreases dramatically, and Figure 6(c) shows that this directly translates to performance degradation when compressing that dimension.
> - **Decrease in SNR with learning rate explains performance degradation of low-memory optimizers at large learning rates**: As the learning rate increases, SNR decreases across layers (Figure 7, Section 4.2). This aligns with the empirical observation that low-memory Adam variants show increasing performance gaps relative to Adam at large learning rates (Figure 1c, Figure 9).

---

> ### Author Response · Authors · 2025-12-01
>
> Thank you for your ongoing support and engagement with our work. We appreciate the opportunity to make these changes more visible. Below, we summarize the major additions in the main text, organized by sections.
>
> **Section 4.2**: We added a summary of batch size and $\beta_2$ analysis, showing that the learning rate remains the dominant factor in reducing SNR if optimal $\beta_2$ is chosen for each batch size. The detailed experiments are provided in Appendix G.2. This addresses the learning rate is the only hyperparameter analyzed concern raised by the reviewer.
>
>
> **Section 5**: We revised most of Section 5 to address concerns regarding the proxy model or memory spikes. During the rebuttal, we have further added a flowchart visualizing the complete SlimAdam method, including when to derive new rules versus reusing existing ones. We believe these changes clarify that our method avoids memory spikes by using a proxy model for SNR analysis, which is supported by empirical evidence that rules generalize from smaller to larger models for a given training task.
>
>
> **Section 3**: We added a summary of our theoretical analysis of SNR in Appendix C.2. We show that both  SNR and the condition number of Adam’s preconditioner measure the dispersion of the second moment distribution around the mean. Therefore, SNR can be thought of as analyzing the condition number of Adam’s preconditioner. During the rebuttal, we have further improved the theoretical foundations of SNR: In Section 3 and Appendix C.1, we show that the relative compression error is equal to 1/(1+SNR). This result directly relates SNR to compression.
>
>
> **Section 5**: We also scaled up our experiments to models with up to 1B parameters (Figure 13), which we summarize in Section 5.
>
>
> We believe these changes to the theoretical grounding in Section 3, the expanded hyperparameter analysis, and the clarified method directly respond to your feedback. However, if there are specific aspects that need further development, we would greatly value your guidance.

---

### Official Review · Reviewer_hoHf · 2025-10-28

**Soundness:** 2
**Presentation:** 3
**Contribution:** 2
**Rating:** 4
**Confidence:** 3

**Summary:**

The paper proposes to compress the second moment tensor of Adam by replacing per coordinate value with the average across specific dimensions. The method defines the signal to noise ratio of the second moment during training and compresses when this ratio is large.

**Strengths:**

As the memory of the optimizer accounts for a significant fraction of the memory requirement for neural network training, compressing it is an important problem. The paper studies a simple method for this task and gives thorough evaluation on different tasks and different modules of the network.

**Weaknesses:**

The method seems to have a large overlap with existing literature. The paper mentions Adam-mini, which already has a large overlap in terms of both the algorithm and the intuition behind the approach. Similar techniques also appear in several other papers such as

Lean and Mean Adaptive Optimization via Subset-Norm and Subspace-Momentum with Convergence Guarantees. ICML 2025

APOLLO: SGD-like Memory, AdamW-level Performance. MLSys 2025

These papers additionally save memory for the momentum, resulting in less memory than the method proposed here. In light of these works in addition to Adam-mini, the contribution of the new paper seems limited.

A minor point: please cite the published versions of the references. For example, the Adam-mini paper is in ICLR 2025.

**Questions:**

The pre-conditioner changes over time as Adam changes V in every step of training. The SNR analysis is fixed up front and the state is compressed in exactly the same way throughout but the condition number of V could change over time. Do you see any change in the condition number of V over time, and if not, is this a property of the training data and are there cases where the condition number of V changes?

In some work, it is mentioned that gradient descent operates on the "edge of stability", do you see any changes if SNR analysis is done at different step sizes?

---

> ### Author Response · Authors · 2025-11-21
>
> We thank the reviewer for the time they invested in assessing our paper. We appreciate that they acknowledge the practical simplicity of our methodology and consider the memory compression problem we study to be important for the field.
>
> > The paper mentions Adam-mini, which already has a large overlap in terms of both the algorithm and the intuition behind the approach.
>
> While both Adam-mini and our work compress second moments, they fundamentally differ in approach and contributions, as we discuss below:
>
> - **Approach differences**: Adam mini uses Hessian-based analysis, requiring expensive eigenvalue computations to determine block partitioning. By comparison, our work introduces a lightweight SNR analysis that can be used to examine compressibility at large scales.
> -  **Different Compression Strategies and Superior Stability**: Compared to Adam-mini, our analysis reveals different optimal compression dimensions for several key layer types (Table 1 and Appendix B). These differences are not arbitrary, as they lead to SlimAdam's superior stability at large learning rates (Figure 1c and Figure 8).
> - **Factors affecting compressibility**: Our analysis reveals that second moment compressibility is not static and it reduces with larger vocabulary sizes, large learning rates, and suboptimal initialization schemes (Section 4). Furthermore, we also show that compressibility heavily varies across training tasks (Section 3).
>
>
> > Similar techniques also appear in several other papers such as [1, 2]. These papers additionally save memory for the momentum, resulting in less memory than the method proposed here.
>
> We thank the reviewer for sharing these works. While we acknowledge that these works save additional memory by compressing the first moments, our work has fundamentally different goals and contributions:
>
> 1. **Different goals and scope**: Our primary goal is to understand when and why Adam’s second moments can be compressed. To this end, we introduce the SNR metric, which quantifies how many distinct second moments Adam actually ‘utilizes’ during training. Our SNR analysis reveals that second-moment compressibility varies systematically across training tasks, architectures, and hyperparameters, providing fundamental insights into Adam's behavior that are not addressed by the references.
> 2. **Interpretable layer-wise insights**: Our SNR analysis reveals that optimal compression strategies vary by layer type (Table 1), and its value quantifies that not all layers are equally compressible.
> 3. **Factors affecting compressibility**: We systematically show that second moment compressibility is not static and it depends on:
>     - Vocabulary size (Section 4.1, Figure 6): Larger vocabularies reduce token embedding compressibility due to heavy-tailed token distributions
>     - Learning rate (Section 4.2, Figure 7): Higher learning rates reduce compressibility
>     - Initialization (Section 4.3, Figure 7c,d): Good initialization (e.g., Mitchell initialization) exhibits higher compressibility.
>     - Training tasks (Section 3): We show that compressibility can vary across training tasks.
>
>
> > A minor point: please cite the published versions of the references. For example, the Adam-mini paper is in ICLR 2025.
>
> Thank you for pointing this out. We have now updated the references to their published variants.
>
> > The pre-conditioner changes over time as Adam changes V in every step of training. The SNR analysis is fixed up front and the state is compressed in exactly the same way throughout but the condition number of V could change over time. Do you see any change in the condition number of V over time, and if not, is this a property of the training data and are there cases where the condition number of V changes?
>
> We thank the reviewer for this question. We clarify that we examine the SNR throughout training (time-averaged SNR, introduced in Section 3.1) to determine the optimal compression dimension for each layer. By averaging over time, we identify dimensions where compression remains safe across the entire optimization process, not just at initialization.
>
> Regarding the condition number, in Appendix C.2, we show that both SNR and the condition number of Adam’s preconditioner matrix quantify the dispersion of the second moment distribution. Therefore, our SNR trajectories (Figures 2, 16-26) can be viewed as analyzing how the condition number of V evolves over time, and we expect to evolve similarly to SNR during training.

---

> > ### Comment · Reviewer_hoHf · 2025-11-25
> > **Acknowledgment**
> >
> > I thank the authors for the detailed responses. The authors answered my questions regarding measuring the SNR averaged over time and how varying the step size also changes the achievable compression. I remain concerned with the overlap with especially Adam-mini and also other prior works. The prior works such as Adam-mini allow for different grouping of coordinates to share parameters and their experiments lead to one particular choice of the grouping. Even though the actual grouping of this paper and Adam-mini might be different (row vs column of the parameter matrix), this seems to be similar instantiations of the same framework to me. The groups need not even be the whole row or the whole column and would still fit in the framework. I consider the contribution to be a slightly more in depth investigation of the same framework, with attention to different parameter types, tasks, etc.
> >
> > In terms of the practical memory saving compared with Adam-mini, the difference is minor because they both compress the second moment term to a very small amount, and the optimizer state is mostly the first order term. Thus, one method might use 51% memory of vanilla Adam and the other might use 52%, which is a very negligible difference. Even with more recent work compressing the first moment, the relative difference is still similar. This work is a reasonable contribution to me but not quite at the level of a top conference so I still put it slightly below the bar.

---

> > > ### Author Response · Authors · 2025-12-01
> > >
> > > We appreciate the reviewer's feedback, but we respectfully disagree that our work is merely an extension of the same framework. Our core contribution is not a new compression scheme, but a framework for understanding when compression is safe. Adam-mini demonstrates that compression can work for language modeling with specific choices. Our work answers the more fundamental question: when and why does compression work, and when does it fail?
> > >
> > > Regarding practical memory saving compared with Adam-mini: Our contribution isn't marginal memory savings, but the reliability of low memory optimizers. The barrier to adopting low-memory optimizers isn't 1% more memory savings but uncertainty about when they'll work. Our work contributes to this understanding.

---

> ### Author Response · Authors · 2025-11-21
>
> > In some work, it is mentioned that gradient descent operates on the "edge of stability", do you see any changes if SNR analysis is done at different step sizes?
>
> Yes, we systematically studied the effect of learning rate on SNR in Section 4.2. We find that SNR decreases with increasing learning rate across all layer types (Figure 7). In Figure 8 (now Figure 9), we show that st small learning rates, about 99% of second moments are compressible, whereas at large learning rates, SNR predicted savings significantly decrease (~30% for GPT pre-training). We also analyzed the effect of batch size in Appendix G.2, and show that SNR trends do not vary extensively with batch size, and a higher learning rate remains the primary factor reducing compressibility.

---

### Official Review · Reviewer_LfWr · 2025-10-29

**Soundness:** 3
**Presentation:** 2
**Contribution:** 3
**Rating:** 4
**Confidence:** 3

**Summary:**

The authors present SlimAdam, a memory-efficient  version of Adam optimizer which achieves up to 99% memory savings by compressing the large second-moment statistics used in Adam's adaptive learning rate computations.  Rather than storing full per-parameter second moments, SlimAdam takes mean of these moments along the fan-in or fan-out dimensions 'when' appropriate. The 'when' in context is guided by a Signal-to-Noise Ratio (SNR) metric.

SNR measures the concentration of second moment values (square of mean/variance). Higher SNR indicates tighter clustering, which justifies compression. And the compression is applied only in the layers where SNR is high, and state granularity is retained where SNR is low. The authors also state that since different layers show compression viability across different dimensions (fan-in/fan-out), derivation of compression rules are required for each model. To determine compression rules for each layer, the authors propose training a small proxy model at a reduced learning rate. SNR statistics from the proxy reliably generalize across larger models of the same architecture and task, informing safe compression dimensions for the full target model.

The authors conduct a compreshensive empirical analysis across a wide range of large models and training tasks, revealing nuanced differences in compressibility across various layer types. Their findings highlight that attention components (such as keys and queries), value and projection layers, MLP layers, and token embedding/vocabulary layers each exhibit distinct compression characteristics. This detailed analysis reveals important, insightful architectural patterns that govern how adaptive moment compressibility varies.

Overall, the paper makes a relevant contribution to efficient optimization for large-scale deep learning, addressing a critical bottleneck in resource consumption. It balances rigorous analysis with practical effectiveness, although clearer exposition, especially regarding the proxy model methodology, would improve accessibility. SlimAdam, hence, is a valuable addition for researchers without sacrificing Adam's effectiveness.

**Strengths:**

1) SlimAdam achieves 99% memory savings compared to the original Adam optimizer, while fully preserving Adam’s effectiveness. It can be seamlessly swapped in place of Adam without requiring any code modifications or additional overhead.

2) The paper presents clear and well-motivated research questions supported by extensive experiments across diverse model architectures and tasks, demonstrating robust generality.

3) The authors provide a detailed algorithmic description alongside publicly available code, ensuring reproducibility.

4) Ablation studies are thoughtfully designed and thoroughly explained, offering valuable insights into the contributions of individual components and hyperparameters.

**Weaknesses:**

1) The main method (SlimAdam algorithm) is explained only in the appendix, and critical implementation insights (proxy model construction, SNR statistics collection) are not clearly presented in the main text. This prevents immediate accessibility and understanding.

2) The concept and practicalities of the proxy model for collecting SNR statistics are not deeply explained. Details about how proxy model size affects SNR relevance and how well proxy-derived rules scale to actual large models could be clearer. Also, how much compute overhead is added for such proxy runs should also be mentioned.

3) The details of SNR statistics adoption over the training steps in the actual model could be explained as well.

Minor Weaknesses:
Appendix C.1 is not completely written.

**Questions:**

1) How does proxy model size affect the SNR statistics for different tasks and architectures?

2) The paper states that for proxy model ignores early SNR statistics, and averages SNR values for next few steps rather than all steps; is the same applied for full model as well- meaning, is compressibility not applied for the first few runs, and how is it adapted over training steps?

I am amenable to changing the score if the questions and weaknesses are addressed.

---

> ### Author Response · Authors · 2025-11-21
>
> We appreciate the reviewer’s thoughtful appraisal of our work. In particular, we are pleased to see that they acknowledge the comprehensive nature of our analysis across models and training settings and that they view our paper as a relevant contribution to the literature on efficient optimization for deep learning. We will now take a moment to discuss each of the concerns raised in the review.
>
> > The main method (SlimAdam algorithm) is explained only in the appendix, and critical implementation insights (proxy model construction, SNR statistics collection) are not clearly presented in the main text. This prevents immediate accessibility and understanding
>
> We thank the reviewer for their suggestion. Significant space in the manuscript is devoted to the presentation of our SNR analysis across training settings and scales because we actually feel this is also a primary contribution of our work; the choice of title for the paper is somewhat indicative of this balance. However, in response to this comment, we have updated the description of the method used to obtain compression rules in Section 5. We also added a diagram that visually illustrates the complete procedure, showing the flow from proxy model → SNR analysis → compression rules → target training.
>
> > The concept and practicalities of the proxy model for collecting SNR statistics are not deeply explained. Details about how proxy model size affects SNR relevance and how well proxy-derived rules scale to actual large models could be clearer. Also, how much compute overhead is added for such proxy runs should also be mentioned.
>
> We believe the revised description of the SlimAdam algorithm directly addresses the SNR statistics and proxy model details.
>
> As for how the model size affects SNR, Appendix J systematically studies how SNR-derived compression rules vary across model sizes and datasets. Specifically, Appendix J.2 compares the SNR-derived compression rules for a 10M parameter model (width=256, 4 layers) against a 100M parameter model (width=768, 12 layers). We find that the depth-averaged SNR yields identical compression rules for these two models, matching those in Table 1. The smaller proxy model in this experiment adds ~10% computational overhead to determine the compression rules. We would like to highlight that once these rules are identified, they can be used to train a larger target model without any additional computational cost.
>
> > The details of SNR statistics adoption over the training steps in the actual model could be explained as well.
>
> >  The paper states that the proxy model ignores early SNR statistics, and averages SNR values for next few steps rather than all steps; is the same applied for full model as well- meaning, is compressibility not applied for the first few runs, and how is it adapted over training steps?
>
> We thank the reviewer for the opportunity to clarify this. We want to clarify an important aspect of SlimAdam: **SNR statistics are used only during the rule extraction phase (Step 2) and are NOT dynamically adapted during target model training (Step 3)**.
>
> The reviewer correctly notes that during proxy model training (Step 1), we ignore early measurements and average SNR over the remaining steps (now added to Section 5). We exclude early steps because Adam's second moments​ require time to accumulate reliable statistics. For target model training (Step 3), we determine the compression rules from proxy SNR statistics, and they remain completely static throughout training. Crucially, no SNR monitoring occurs during target training, compression is applied from the first training step, and there is no dynamic adaptation over training steps.
>
> > Minor Weaknesses: Appendix C.1 is not completely written.
>
> We thank the reviewer for pointing out the incomplete statement. In response, we have updated this sentence.

---

### Official Review · Reviewer_75Xb · 2025-10-31

**Soundness:** 2
**Presentation:** 2
**Contribution:** 1
**Rating:** 2
**Confidence:** 4

**Summary:**

This paper proposes a metric called SNR (gradient version) to address the high memory usage of Adam’s second momentum. The paper compresses the second-momentum tensor along these dimensions to a single mean value (or small number of values). The authors claim this approach (SlimAdam) can save up to 99% of the second-moment memory while maintaining Adam’s property and stability.

**Strengths:**

•	The idea of quantifying the compressibility of Adam's second moments on a per-layer, per-dimension basis is interesting.

•	The authors conducted extensive experiments across various architectures (GPT, ViT, ResNet) and tasks.

**Weaknesses:**

1.	Lack of Theoretical Foundation: As an optimizer paper, it lacks a convergence proof and relies almost entirely on experimental observations (e.g., using a 10x lower LR).

2.	Missing Essential Data: The paper does not include 'loss vs. step' curves, a critical metric for evaluating optimizers.

3.	Methodological Ambiguity: There is no logical basis for the proxy model design or the SNR threshold of $\alpha=1$.

4.	Questionable SNR Justification: The underlying assumption of mapping the mean of $V_t$ to 'signal' and its variance to 'noise' is not justified. (Since the justification of SNR relates to original vs. noise)

5.	Exaggerated Contribution & Poor Comparison: The 99% claim is misleading (it's 50% of the total), and comparisons to SOTA optimizers that compress both moments (e.g., SMMF: Square-Matricized Momentum Factorization for Memory-Efficient Optimization which reduces the first and second momentums so that the total compression ratio is by up to 96%) are missing.

**Questions:**

•	What is the theoretical justification for treating the mean as 'signal' and the variance as 'noise' in your SNR definition $SNR_K = \mathbb{E}[(\mathbb{E}_K[V_t])^2 / Var_K[V_t]]$ from an optimization perspective? Is there a theoretical basis to claim that a low-variance (high SNR) tensor is inherently 'compressible'?

•	What is the specific theoretical justification for choosing the SNR threshold $\alpha=1$? Can you guarantee this value is universally optimal across different tasks and model architectures?

•	Can you provide loss-vs-step curves for your key results (e.g., Figure 8) to demonstrate that SlimAdam achieves the same 'final' performance with the same convergence speed as Adam?

•	Compared to optimizers like SMMF (2025), which compress both first and second moments for a 96% total memory saving, what is the practical advantage of SlimAdam, which only compresses the second moment for a ~50% total saving?

---

> ### Author Response · Authors · 2025-11-21
>
> We appreciate your time in reviewing our work and providing comments and feedback. Below, we address individual questions and comments.
>
> > Theoretical foundations of SNR
>
> While we do not provide a formal convergence proof, we theoretically analyze SNR as a metric (summarized in Section 3):
> - In Appendix C.1, we have now established that the relative L2 compression error incurred on replacing the second moments with their mean is given by $1 / (1 + SNR)$. Hence, high SNR indicates the mean provides a good approximation, while low SNR suggests significant information is lost during compression.
> - In Section C.2, we relate SNR with the condition number of Adam’s preconditioner $\kappa = \min_i \sqrt{v_i} / \max_i \sqrt{v_i}$. Both of these metrics quantify the dispersion of the second moment distribution, establishing that SNR analysis can be viewed as examining Adam’s preconditioner.
> - In Section C.3, we prove that SNR = 1/2 for iid Gaussian distributed gradients. This result confirms that SNR is independent of the scale of the gradients, suggesting that a single cutoff could work well in practice, which we then empirically validate (Figure 9).
>
> We also note that many well-known works, such as Lion and Adafactor, do not provide convergence proofs in their original paper and provide extensive empirical validation.
>
> > Using 10x smaller LR for SNR analysis
>
> We take the opportunity to clarify that while our prescription of performing SNR analysis at a 10x smaller learning rate is based on empirical evidence, it is not arbitrary. The motivation behind this choice is that compression is maximized at small learning rates (Figure 9). Here, the intuition is that using a small learning rate avoids artifacts that emerge when training Adam at large learning rates, and captures the latent fundamental compression rules that the problem admits.
>
> > Missing loss-vs-step curves
>
> We are not sure why the reviewer asserts that the paper is missing “loss vs. step” curves. In Figure 1(b, c) of the submission copy, we provide the training loss vs. step curves for the GPT pre-training task. This result shows that while all low memory Adam variants follow Adam’s curve throughout training at small learning rates, at large learning rates, SlimAdam exhibits nearly the same training dynamics as Adam, while other variants experience training instabilities. That said, in the service of completeness, we have also added loss vs. step curves for all Figure 8 (now Figure 9) experimental configurations in Appendix L.
>
> > What is the specific theoretical justification for choosing the SNR threshold $\alpha=1$? Can you guarantee this value is universally optimal across different tasks and model architectures?
>
> While we cannot provide a theoretical guarantee of universal optimality, the combination of SNR being scale invariant and extensive empirical validation across diverse tasks makes $\alpha = 1$ a principled and practical choice.
>
> **Theoretical motivation:** In Section C.3, we prove that for iid Gaussian gradients, SNR=½ and does not depend on the matrix dimensions or the scale of the gradients. This scale invariance is a necessary condition for a universal threshold to exist across layers, tasks, and architectures. If our metric depended on dimensions or the gradient scale, we would need different thresholds.
>
> **Empirical validation:** Our empirical choice of SNR = 1 is more conservative than the Gaussian gradient analysis, and it stems from the simple intuition that when SNR > 1, the standard deviation is smaller than the mean, indicating that entries are clustered around their mean. In this regime, replacing individual second moments with their shared mean would incur a small error. Our experiments shown in Figure 9 validate that the choice of SNR = 1 works without any tuning across training tasks: GPT pre-training, Llama finetuning, ResNet, and ViT image classification.
>
> > Proxy model justification
>
> The proxy model approach addresses a fundamental practical challenge: SNR analysis requires sufficient memory to run full Adam to collect second moment statistics to derive the compression rules, which may not be available in resource-constrained settings. Our solution is to perform SNR analysis on a smaller proxy model that fits in memory with full Adam, and then transfer the compression rules to the larger target model. The approach is based on the empirical finding that compression rules derived from a smaller proxy model (depth = 4, width=256) yields similar rules to that of a larger model (depth = 24, width = 1024), but with small variations as shown in Appendix J.2. The small variations disappear when using depth-averaged SNR and result in the rules summarized in Table 1 and discussed in Section 5. We have now improved the description of SlimAdam's three-step process and also added a flowchart visualizing the complete SlimAdam method, including when to derive new rules versus reusing existing ones.

---

> ### Author Response · Authors · 2025-11-21
>
> > The 99% claim is misleading (it's 50% of the total)
>
> **Our claim of "saving up to 99% of total second moments" is accurate and clearly stated throughout the paper**. We consistently report savings relative to Adam's second moment memory, not total optimizer memory. For instance:
> - Abstract: "saving up to 99% of total second moments"
> - Section 5: "SlimAdam saves 99% of second moments in GPT pre-training"
> - Figure 9 caption: "Fraction of reducible second moments (relative to Adam)"
>
> In terms of total optimizer memory (first + second moments), SlimAdam saves approximately 50% for GPT pre-training. We do not claim 99% total optimizer memory savings. To avoid any misinterpretation, we now also mention 50% total memory savings throughout the work.
>
> > Comparison with SMMF and our Contributions
>
> We thank the reviewer for pointing out SMMF. However, **our work has fundamentally different goals and contributions**. Our primary goal is to understand when Adam’s second moments can be compressed. Our analysis yields:
> 1. **Interpretable layer-wise insights**: Our SNR analysis reveals which layer types are compressible (e.g., attention projections, MLP.Down) vs. incompressible (e.g., token embeddings along vocabulary dimension, attention keys/queries along head dimension). This layer-specific understanding is summarized in Table 1.
> 2. **Factors affecting compressibility**: We systematically show that second moment compressibility is not static and it depends on:
>  - Learning rate (Section 4.2, Figure 7): Higher learning rates reduce compressibility
>  - Vocabulary size (Section 4.1, Figure 6): Larger vocabularies reduce token embedding compressibility due to heavy-tailed token distributions
>  - Initialization (Section 4.3, Figure 7c,d): Better initialization (e.g., Mitchell) increases compressibility.
>  - Training task (Section 3 and Table 1)
>
> While SMMF uses matrix factorization to compress both moments for maximum memory savings, our work provides an interpretable analysis of when second-moment compression is safe and which dimensions to compress per layer type. These approaches could potentially be combined. We will add SMMF to the related work discussion.

---

### Author Response · Authors · 2025-11-21

We thank all the reviewers for taking the time to review our work and providing valuable feedback. We have significantly updated our submission to incorporate your suggestions. Below, we summarize the main changes.

- **SNR and relative compression error relationship (Section 3 and Appendix C.1)**: We have established the relationship between SNR and the relative L2 compression error. This provides a theoretical justification for using SNR as a measure of compression -- higher SNR results in lower relative compression error, whereas lower SNR suggests significant information is lost during compression.
- **Training dynamics (Appendix L)**: We have now added comprehensive loss and accuracy trajectories for all experiments in Section 5 and Figure 8 (now Figure 9), demonstrating that SlimAdam matches Adam's convergence speed and final performance throughout training across all tasks (GPT pre-training, Llama fine-tuning, ResNet, ViT).
- **Improved SlimAdam algorithm description (Section 5)**:  We have significantly improved the description of SlimAdam's three-step process (SNR collection, rule extraction, deployment) and clarified how practitioners can reuse existing rules (Table 1) for known tasks. We have also added a flowchart visualizing the complete SlimAdam method, including when to derive new rules versus reusing existing ones.

---

### Author Response · Authors · 2025-12-01
**Final Author Remarks**

We thank all the reviewers, originally assigned AC, and newly assigned AC, for their efforts throughout the review process. We especially thank the reviewers for their comments and suggestions. We genuinely believe that this discussion has helped us improve the paper. Below, we summarize the major improvements and clarifications for completeness.

**Improved theoretical foundations of SNR**: We have now established that the relative L2 compression error incurred on replacing the second moments with their mean is given by 1/(1+SNR), as described in Section 3 and Appendix C.1. Hence, high SNR indicates the mean provides a good approximation, while low SNR suggests significant information is lost during compression.

**Improved description of the SlimAdam method**: We have now improved the description of SlimAdam's three-step process and also added a flowchart visualizing the complete SlimAdam method, including when to derive new rules versus reusing existing ones.

**Contribution clarifications**: We clarify that our goal is to understand when and why Adam’s second moments can be compressed throughout training across training tasks. To this end, we introduce and analyze SNR, which directly measures the relative compression error when the second moments are replaced by their mean along row/column. Our SNR analysis reveals that second-moment compressibility varies systematically across training tasks, architectures (Section 3), and hyperparameters (Section 4), providing fundamental insights into Adam's behavior that are not addressed by the referenced work.

---

### Meta-Review · Area_Chair_ueoS · 2025-12-20

**Summary:**

The paper proposes a new method that compresses Adam's second-moment memory by replacing many per-parameter values with a few dimension-wise averages when a layer looks "compressible" as judged by an SNR score.

The reviewers highlight the potential practical aspect of the method as well as the empirical study across several architectures (Llama, ViT, ResNet) and tasks.

However, the reviewers also raised major criticisms such as limited theoretical grounding and several heuristic choices (proxy-model design, reduced learning-rate protocol for SNR collection, and the SNR cutoff) that initially felt under-justified. Multiple reviewers also question novelty, emphasizing overlap with Adam-mini and other recent memory-efficient optimizers. I do feel this part of the paper requires significant improvements. After some exchanges with the reviewers, my own understanding is that both methods do share a lot in common. The authors claim that Adam-mini focuses on the use of the Hessian but I do not think this is an accurate reading of the Adam-mini paper. I would say that Adam-mini's structure is Hessian-block-driven whereas SlimAdam's structure is SNR-driven.

Another repeated concern is practicality: choosing compression axes and rules may require extra runs or proxies, and if full-size warm-up were required it would negate the memory benefit.

I believe the authors did a good job in the rebuttal, addressing many concerns, but it feels that the paper would benefit from the following improvements:
1) Include a concise comparison table vs Adam-mini, SMMF, APOLLO, etc
2) Clarify SNR vs simpler metrics with a minimal ablation
3) Several heuristics raised reviewer's skepticism, I think these should be better justified
4) Minor but consider toning down some of the claims (99% claim especially)

Overall, I think this paper is not quite ready for publication, but I do think the direction is promising and I think a revision of the paper will be a strong submission for an upcoming conference.

**Reviewer Concerns:**

Please see meta-review where I addressed this as best as I could.

**Reviewer Scores:**

There were some discussions with the reviewers before the discussions were frozen, but the reviewers did not significantly change their scores.

---

### Decision · Program_Chairs · 2026-01-26

Reject